# OSINT-Based LPC-MTD and HS-Decoy for Organizational Defensive Deception

**Sang Seo and Dohoon Kim \***

Department of Computer Science, Kyonggi University, Suwon-si, Gyeonggi-do 16227, Korea; tjtkd8271@kyonggi.ac.kr
\* Correspondence: karmy01@kyonggi.ac.kr

**Abstract:** This study aimed to alleviate the theoretical limitations of existing moving target defense (MTD) and decoy concepts and improve the efficiency of defensive deception technology within an organization. We present the concept of an open-source intelligence (OSINT)-based hierarchical social engineering decoy (HS-Decoy) strategy while considering the actual fingerprint of each organization. In addition, we propose a loosely proactive control-based MTD strategy that is based on the intended competitive exposure of OSINT between defenders and attackers. Existing MTDs and decoys are biased toward proactive prevention, in that they only perform structural mutation-based attack avoidance or induce static traps. They also have practical limitations, e.g., they do not consider security characterization of each organizational social engineering attack and related utilization plans, no quantitative deception modeling is performed for the attenuation of the attack surface through exposure to OSINT, and there is no operational plan for optimal MTD and decoy application within the organization. Through the applied deception concepts proposed here, the total attack efficiency was reduced by 287% compared to the existing MTD and decoys, while the artificial deception efficiency dominated by defenders was improved by 382%. In addition, the increase rate of deception overhead was also reduced by 174%, and an optimized deceptive trade-off was also presented. In order to enable an organization to utilize the OSINT concept, statistical error reduction, and MTD mutation cycle-based deceptive selectivity, it was introduced as a loose adaptive mutation rather than a preferential avoidance strategy, and an organization-specific optimization direction was introduced through a combination of HS-Decoy and LPC-MTD. In the future, in order to improve the operational reliability of the HS-Decoy and LPC-MTD-based combined model and standardize threat information for each organization, we intend to advance it into an international standard-based complex architecture and characterize it as game theory.

**Keywords:** cyber deception; moving target defense; decoy; open-source intelligence; social engineering

## 1. Introduction

Cyber deception [1] is a deception technology that lures and deceives an attacker who gains access to a specific system or network. Its main purpose is to deceive an attacker by using a deceptive intended security vulnerability as bait to lure the attacker, which results in unique defense mechanisms (e.g., induction, isolation, backtracking, and mutation) that are different from other conventional security elements and an active deceptive kill chain dedicated to each scenario [2,3]. In other words, the introduction of cyber deception makes it easy to build an adaptive security system specializing in information collection and active defense. This is accomplished by mitigating the temporal and spatial inequality relationship that attackers have in existing conventional security environments, and by dynamically detecting and responding to the attack vector or intrusion path of the attacker. It can also be applied as a source technology for the concept of active defense to realize the following: artifact collection for backtracking, specialized induction and isolation for social engineering threats, immediate detection and proactive avoidance of advanced APT, and the reduction of "entropical" capture time for intrusion attempts.

### 1.1. Background and Limitations of Existing Cyber Deception

The detailed element technologies of cyber deception include moving target defense (MTD), Honey-X, and decoys. In particular, the most efficient of these is network-based MTD (N-MTD) [4] that functions by increasing the network attack and exploration surface [5] through actions, such as shuffling, diversity, and redundancy, which focus on the fluid properties of the network [6]. This defense is realized proactively by periodically or aperiodically mutating defender information that may be the target of cyberattack according to the intention of the defender. Further, this assists in avoiding attack attempts, mitigating the asymmetry dominated by attackers, and limiting the effectiveness and chainability of exploitation while using previously collected information. In addition, a decoy is a dynamic sandboxing element that performs partial induction and isolation by encouraging an attacker to attack a false target. Researchers have studied N-MTD in combination with various decision strategies and theories, including game theory [7,8], Markov decision [9,10], genetic algorithm [11], and machine learning [12], to gain certain advantages. These include formulating the deception threshold while reducing the initial invasion influence and surface in the cyber kill chain [13], diversifying the optimal mutation strategy, and even performing a mutual combination between Honey-X and decoys. However, these previous studies that are related to existing deception imply the following theoretical limitations.

- The existing MTD only performs structural mutation-based avoidance or indirect induction to static traps, resulting in the limitation of focusing on securing only a proactive defense. In addition, functional defense for post-response was not considered as a major factor, leading to no countermeasures when sophisticated attackers, who identify MTD patterns or vulnerable attack points, bypass the system.
- It is necessary to construct a dedicated deception model to apply MTD and decoys to each organization's operating environments and cater for organizational uniqueness. This is also related to seamless connection [14,15] in the case of an MTD mutation. However, previous studies have reported partial mutation after opening a separate encrypted closed-communication channel and mapping random virtual network information with real information, which results in low actual operability. In addition, an attacker may be able to penetrate the defense when virtual information is randomly configured with characteristics disparate to that of the organization's.
- It is necessary to alleviate detection in the organization, skewness, and outliers within the blocking system, and also consider the following: MTD mutation that is related to social engineering factors, such as open-source intelligence (OSINT) [16,17] and human information (HUMINT) in organizations, decoy management, and dynamic diversification of the attack and exploration surface.
- The deception efficiency of MTD and decoys against malvertising [18], spear phishing [19], and watering hole-based [20] social engineering attacks [21] should be considered. In other words, OSINT-based modeling for MTD and decoys should be quantified according to the organizational legacy environment. This may refer to the statistics subject to frequent social engineering infringements by APT, as the organizational information of major countries, including the United States and the United Kingdom, is disclosed through third party-based OSINT [22,23].
- There are many approaches to reduce the statistical skewness and errors of threat information or to improve the response quality of the organizational platform by using a research model that integrates the existing MTD and decoys. However, these approaches are limited to the configuration of an experimental environment that is based on a specialized domain. In addition, the configuration of the deception policy for each public OSINT was not considered, except for a few vendors (e.g., Illusive Networks [24]), who focused on artificial disinformation and the contamination of external OSINTs, rather than the collection of internal OSINTs.

*1.2. Research Goal and Major Contributions*

To optimize the existing MTD and decoy's operations for each organization and improve the deception efficiency based on OSINT in social engineering, this study first attempts to calculate a hierarchical social engineering decoy (HS-Decoy) that is based on actual OSINT characteristics. We also propose a loosely proactive control-based MTD (LPC-MTD), which deliberately exposes the attack points by reducing the skewness and noise of the existing MTD but reflecting the high-level attack in the sandbox as it is to construct high-quality artifacts for sophisticated attackers. We performed organization-specific simulations based on cyber kill chain and APT scenarios after combining the LPC-MTD's adaptive mutation cycle selection scheme and HS-Decoy's OSINT-based disinformation, derivation, and isolation scheme. Subsequently, deception sequences, complex experimental metrics, and the optimal trade-off value were calculated. The following are the major contributions of this study.

- By applying social engineering OSINT and loose mutation concepts in MTD and decoys, it was possible to secure a reinforcement plan for the proactive prevention and post-response in an organizational environment, and also improve the optimized deception attack-defense efficiency for organizational disclosed OSINT's uniqueness.
- It was possible to construct an OSINT-based decoy that could improve both attraction and isolation due to the initial cognitive skewness of the attacker and the persistence efficiency of deception by layering and normalizing the organizational OSINT based on domain name system (DNS) enumeration into entity and link-based graph.
- By accepting attacker infringement attempts and attack continuation as a loose adaptive mutation rather than using a strong preliminary avoidance, it was possible to construct an operational security-based MTD deception methodology that could advance the defender's dominant deception behavior for MTD mutation cycle selection.
- Through the proposed HS-Decoy and LPC-MTD combined modeling, it was also possible to establish a trade-off-based deception optimization strategy that was specialized for each organization, and to enhance it for each organizational strategy.

The remainder of this paper is organized, as follows. Section 2 analyzes previous studies related to MTD and decoys, which are the basic deception concepts in this study. Section 3 describes the OSINT-based HS-Decoy and LPC-MTD concepts and architectures. We provide metrics and equations that ease the limitations of existing MTD and decoys and characterize them by organization. In addition, we discuss the structure and process to combine the two deception concepts, the social engineering OSINT module and configuration of the specialized deception parameters. In Section 4, we compare, analyze, and verify the proposed model through simulation by configuring a software defined network (SDN)-based test bed based on the proposed LPC-MTD and HS-Decoy architectures. Section 5 discusses the findings of the study. Finally, Section 6 presents the conclusions.

## 2. Related Studies and Taxonomy of Proposed Deception Models

We reviewed various previous studies for the composition and analysis of the proposed social engineering OSINT-based HS-Decoy and LPC-MTD that focus on MTD based on periodic or aperiodic mutations. The hierarchical classification of these is network-based MTD, platform-based MTD, software-based MTD, and data-based MTD. Among these, N-MTD, which increases the attack cost, effort, and load of the attacker by disturbing the initial reconnaissance stage by hiding, moving, and attenuating the defender's surface, is the most efficient. Additionally, it can be combined with host-based MTD (H-MTD) to yield significant precautions. Accordingly, the scope of previous studies reviewed to achieve our goal was game theory-based N-MTD, machine learning-based N-MTD, and detailed mutation parameter-based N-MTD. The knowledge that was gained from these was applied to improve the performance of the proposed model.

### 2.1. Game Theory MTD Research

Studies on N-MTD based on game theory focus on modeling the mutation of the defender on a competitive strategy toward the attacker and optimizing the MTD variables for each scenario. The main goal is to maximize the gains of defenders and minimize the gains of attackers by advancing the deception strategy. They are mainly classified into general game theory-based research, Bayesian Stackelberg game theory-based research, and stochastic game theory-based research [5,25,26].

First, in general game theory-based research, Zhu et al. [27] calculated a quantitative MTD trade-off between the security of the defender enhanced by MTD and degraded performance (e.g., service unavailability and increased system reconfiguration cost) with a focus on the correlations and trade-offs specialized for each scenario by applying common game theory equations and metrics to the MTD mutation concept. This was the first time a specialized game theory was established to calculate an MTD applicational strategy for threat backtracking and threat hunting system construction. To ensure high network visibility and throughput and improve cyber agility [28], Ge et al. [29] proposed an incentive-compatible MTD game theory based on a server's migration and communication mapping mechanism between normal users. This introduced the concept of cyber agility in the construct of N-MTD. Carter et al. [30,31] applied game theory as an optimal migration strategy to avoid suspicion of induced and isolated attackers and ensure seamless connection for each legitimate user. This made it possible for them to quantitatively verify that the diversity of cloned platforms for the application of H-MTD at a lower level is more effective for some continuous attacks than pure random selection mutations. However, it was inefficient in performing rapid local attack-related mutations. It was also found that ensuring high hierarchical diversity for each dummy platform led to higher proactive advantages than purely possessing a large number of dummy platforms.

Second, in Bayesian Stackelberg game theory-based research, Hasan et al. [32] proposed CAMP, a Nash equilibrium-based game model to minimize the influence of internal and external threats that are caused by the infringed co-virtual machine (co-VM) while detecting a co-resident attack, a side-channel attack within a virtual environment that shares the same temporal/spatial resources. Through this, it was proved that the proposed Monte Carlo-based CAMP could immediately provide an active attacker induction and isolation tactic related to the optimal MTD defense strategy in a virtual environment by cloning a normal server. Feng et al. [33] proposed an artificial information disclosure model that was based on Bayesian Stackelberg to improve proactive prevention in defender MTD. Based on a signaling game, which is an interactive decision strategy, the defender's intentional disclosure of information would perturb and skew the attacker's initial decision making, proving that it is a promising methodology that could not only reduce attack persistence, but also improve the overall agility of the defender. To apply the concept of artificial information exposure in practical environments, Zhu et al. [34] proposed a Stackelberg game solution that generated and disclosed false packets that aimed at the reconnaissance stage of internal and external attackers, making it possible to improve the efficiency of indeterminate attacker induction and isolation mechanisms in multi-routing networks, and it also improved the practical applicability of the deception concept.

Third, in stochastic game theory-based research, Manadhata et al. [35] proposed an optimal MTD game model based on a mechanical surface by mathematically calculating the strategies for each MTD mutation state reflecting the diversity of the attack-defense dynamics based on probabilistic conversion as an extension of the previous attack surface quantification research [36]. This made it possible to quantify the mutation threshold for each scenario, the attenuation of surface by organization, and MTD equilibrium for modeling the optimal proactive strategy based on environmental characteristics.

### 2.2. Machine Learning MTD Research

The core of machine learning-based N-MTD research is to optimize the MTD using learning techniques, like k-means clustering, reinforcement learning, and Stackelberg-

based deep learning, in order to formalize the defender dominant deception gradient. This includes Lucas et al.'s [37] genetic algorithm-based N-MTD research for the calculation of defender dominant cyber resilience [38] based on mutant evolution reputation.

Colbaugh et al. [39] proposed a prediction-oriented MTD mechanism based on machine learning as an active defense to minimize the chain of adaptive gathering behavior and the negative influence of social engineering APT that bypasses and reverses the naïve MTD strategy that is applied within organizations. The objectives of this research were to construct scenarios based on the attacker sequentially performing stealth, hidden reconnaissance, and indirect firepower projection to overcome the MTD strategy, and show that it improved the efficiency and robustness at the practical level of each organization by performing quantitative evaluation using spam and non-spam email datasets. This made it possible to derive an optimal MTD defense strategy that was difficult for an attacker to reverse engineer based on the co-evolutionary relationship between attack and defense when applying MTD within the organization and applying it as a basic axiom to critical systems.

Sengupta et al. [40] proposed MTDeep as a proactive and applied defense against adversarial machine learning by attackers targeting machine learning environments in organizations. It is a framework that guarantees structural security and input/output robustness of deep neural networks that are configured in networks and hosts. Through this, it was possible to alleviate the asymmetric problem between adversarial machine learning of the attacker and the trained ensemble network through structural redundancy and diversity given by each candidate network in the ensemble. Further, the MTD strategy could be optimized using hyperparameters, providing robustness against internal and external attacks. Farchi et al. [41] also demonstrated the promise of machine learning-based MTD by presenting a strategic learning algorithm selection mechanism to proactively defend against side-channel attacks and sample injection for anti-machine learning and advancing the strategic MTD activation process that is based on the existing game theory.

### 2.3. MTD Research Based on Calculation of Detailed Mutation Parameters

Research on the calculation of detailed mutation parameters for N-MTDs focuses on the design of parameters related to MTD mutation and decoy-specific incentive strategies to accommodate the legacy environments of organizations as opposed to SDN. The main goal of this type of research is to expand the domain knowledge to prevent MTD and decoys from being limited to academic research and allow practical applications to secure high reliability within the organizational environment.

SSCM [42] was proposed to solve the migration scalability problem that was experienced by DESIR [15] in previous research. Under the premise that the host performs network trait mutations such as IP and port changes on a virtualized clone image, it was shown that the routing substrate guarantees transmission and reception of packets using the newly allocated network characteristics and the previous network characteristics even when expired. This alleviated the issues experienced by DESIR where service interruption occurred due to the migration of legitimate servers, and materialized seamless connection for legacy network. HTN [43] was proposed to perform the network trait mutation of legitimate servers that are designated as protection targets within one second in legacy networks of organizations. In addition to guaranteeing the rapid mutation of each legitimate server, it was possible to calculate a method to secure the independence of each module when configuring the decoy-bed and improve the customization of arbitrary address sets to be protected.

### 2.4. Analysis by Previous Studies for Construction of Proposed Deception Model

The above-described studies applied MTD and decoys using hypotheses within restricted scenarios. Therefore, substantial construction issues exist for securing reliability and efficacy of MTD in practical organizational environments, and the enhancement of the association between OSINT-based social engineering decoys remain as a limitation. Therefore, we propose a deception model that combines an OSINT-based HS-Decoy and

LPC-MTD based on artificial information disclosure to alleviate these limitations and improve the applicability and security of deception methods for organizations. Table 1 presents the taxonomy analysis of previous studies and the proposed model.

**Table 1.** Taxonomy of existing moving target defense (MTD) and decoy-based deception research and proposed scheme.

| Approach | Specific Technique | Advantages | Disadvantages |
|---|---|---|---|
| Game theory-based MTD [27–35] | General game theory, Bayesian Stackelberg game, and stochastic game theory | Determining the optimal MTD strategy and realizing the decision model. | Requires additional reasonability and reliability of the calculated decision. Consideration of statistical errors such as stochastic outliers and bias. |
| Machine learning-based MTD [39–41] | Classification or clustering, reinforcement learning, and deep neural network. | Calculating specialized defense indicators for each organization through the introduction of advanced learning concepts. | Requires training with large datasets. Complexity due to computational overhead. Limitations in realization of multi-factor-based deception process. |
| Calculation of detailed mutation parameter-based MTD [42,43] | Before definition of MTD and after definition of MTD. | Realization of visibility and efficacy of practical application of any proposed MTD scheme. | Evaluation dependent on existing simulation models. Limited in performing abstraction in arbitrary legacy nets. Limitations of applicability in real world. |
| **Our proposed deception model** | | Description and Improvement | |
| | **OSINT-based HS-Decoy** OSINT based on DNS enumeration and clustering used correlation metric. Knowledge-based graphDB. Perturbation for configuration of attackers' misperception and decoy properties. | An artificial attack vector composed of fake organizational information based on hierarchical OSINT of network, host, service, platform, and user. The possi-bility of an attacker's initial attempt is improved, and the attacker's suspicion is minimized compared to other decoys as the real characteristics of each organi-zation is used. It proposes a strategy-selective unique deception concept that initially biases and induces the actual attacker's social engineering choice. Through this concept, the limitations of the abstraction model of deceptive simulation and applicability within the organization environments of existing MTD studies [42,43] were mitigated, and proactive efficiency and scalability of decoy were also improved. Mitigation of the limitations of other previous studies [27–35,39–41] was also achieved through the combination with LPC-MTD. | |
| | **LPC-MTD** Loose mutation specialized to each organization. Periodic or aperiodic calculation of OSINT-based mutation targets. Configuration of OSINT shuffling and diversity trade-off within each organization. | An MTD strategy-selective concept that intentionally induces an attempt, then bypasses and reverses the mutation patterns of social engineering attackers, and avoids automated attacks that may cause statistical noise. It is possible to secure all thresholds, redundancy, and diversity related to mutation and mobility, and to consider trade-off-based efficiency by selecting MTD strategies specialized for each organizational environment. Through this concept, the problem of increasing statistical errors and computational overhead presented in the existing MTD studies [27–35,39–41] has been mitigated, and the post-response performance and deceptive resilience of MTD have also been improved. The easing of limitations to other previous studies [42,43] was also achieved through the combination with OSINT-based HS-Decoy. | |

## 3. Proposed Deception Models with Organizational OSINT, LPC-MTD and HS-Decoy

The OSINT-based HS-Decoy and LPC-MTD model presented in this study was designed to alleviate the limitations of existing MTD and decoys that are applied in organizations. Therefore, in this section, we discuss the construction of the proposed organizational OSINT and each concept, module, process, and then present related metrics.

### 3.1. Organizational Social Engineering Deception in the Proposed Model

Figure 1 shows the standardized main overview of each applicational deception concept, i.e., organizational OSINT, LPC-MTD, and HS-Decoy, in the proposed deception model. First, the organizational OSINT and related deception knowledge were standardized through the asynchronous parallel chain-based OSINT process, followed by the application of the knowledge to the applicable concepts, such as LPC-MTD and HS-Decoy.

Subsequently, the integrated OSINT-based deceptive architecture (IODA) was built and integrated based on the configured LPC-MTD and HS-Decoy. The annotations related to equations and algorithms for each concept that will be presented in the remainder of this section can be viewed in Table A7 in Appendix A.

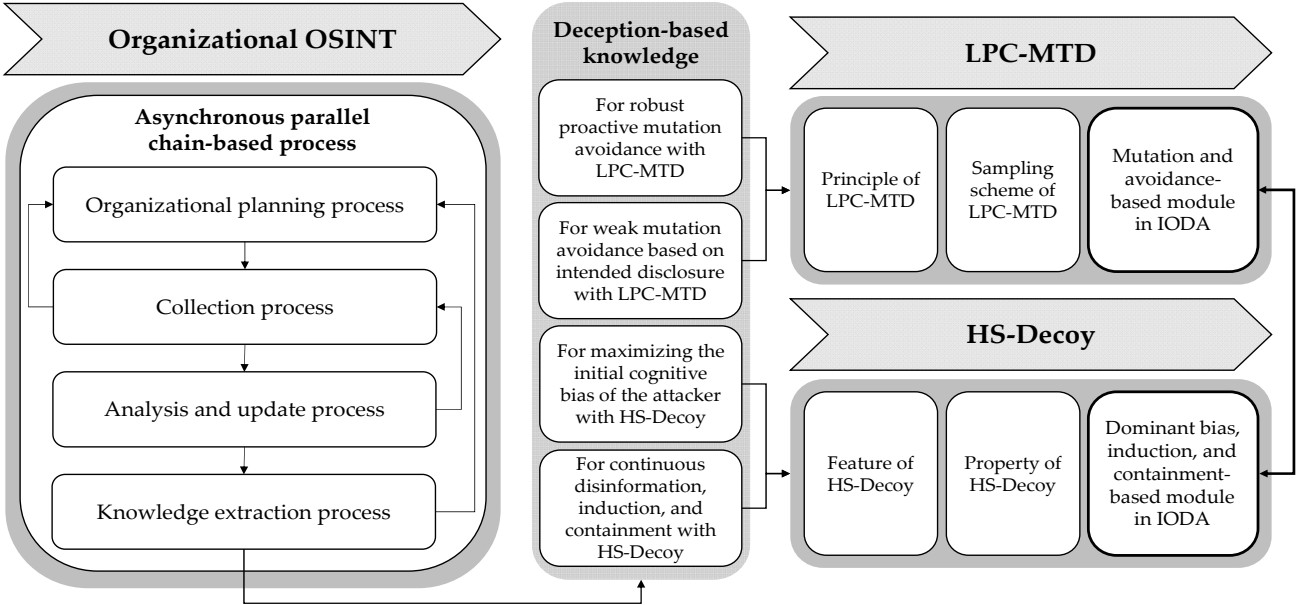

**Figure 1.** Main overview of the proposed open-source intelligence (OSINT)-based deception model with organizational OSINT, loosely proactive control-based MTD (LPC-MTD), and hierarchical social engineering decoy (HS-Decoy).

### 3.1.1. Organizational OSINT for MTD and Decoy

OSINT is a public and reliable source of information from a government agency or other providers. It collectively refers to the process of collecting information from public sources, such as detailed organizational information and unique characteristics. OSINT is used in various fields, such as the military, security, law enforcement, business, and cyber security. In addition, it can be classified as an information flow category that is based on multiple domains, such as media, internet, and government data.

Figure 2 shows the basic OSINT process in the proposed model that focus on the following detailed processes: the organizational planning process that identifies sources of information and establishes the OSINT strategy and security requirements that are related to the organization's legacy operating environment. The collection process during which information is actively or passively collected from public information groups according to the identified organizational information sources and OSINT strategy. The analysis and update process is when acquired data are normalized based on correlation and then reinforced and updated according to its purpose. Finally, the knowledge extraction process involves the organizational OSINT information being advanced through deception-related knowledge to LPC-MTD and HS-Decoy and simultaneously purifying it to create final organizational information. Further, the model is standardized for each organization for the layering and systematization of OSINT and implemented to be configured as an element of crowd operational security with deception and precision to organizational cyber space intelligence (CYBINT).

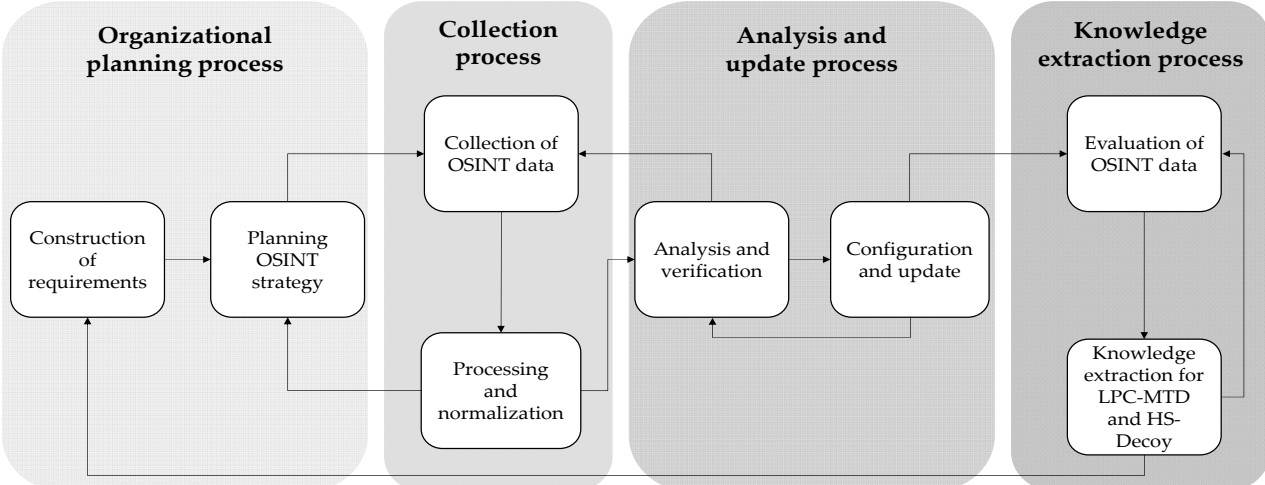

**Figure 2.** Lower-level overview of the asynchronous parallel chain-based organizational OSINT process.

Organizational OSINT information is collected passively, focusing on the asynchronous chain-based OSINT process presented in Figure 2, and then normalized by element and standardized in graphDB to ensure higher speed, adaptability, and responsiveness when compared to existing sequential processes. Subsequently, the decoy feature is reinforced to standardize it, even as a cooperative deception process within a distributed multi-HS-Decoy, and updates are performed on a regular basis to reflect the real-time changes in OSINT trends. At this time, the OSINT group is created, distributed, and updated based on the organizational environment in which MTD and decoy-based deception are applied. To artificially induce the defender's asymmetric superiority relationship and extreme cognitive skewness, reputation is calculated based on the third party referenced during collection, and updated weights for each entity are then applied to attenuate the ambiguousness of the existing OSINT. Thus, the final standardized OSINT information group by major government organizations is constructed in detail, as shown in Table A1 and Figure A1 in Appendix A. The total number of OSINT elements was 26 million. To comply with the research ethics, we calculated and reconstructed the mutation weights hierarchically, using only published fingerprints.

### 3.1.2. LPC-MTD

Existing MTD and N-MTD have been studied to avoid invasion attempts proactively or to improve metrics and variables with a focus on the attenuation of the defender's diversified attack and exploration surface, and then apply it in applied domains, such as IoT [44], cloud [45], and cyber physical systems [46]. Our reason for selecting an MTD mutation cycle using LPC-MTD was to ease the operational deceptive limitations of existing MTD, and to enhance the lack of reactive responses. Further, the aim was to intentionally expose false information or disinform attackers who analyze social engineering APT or MTD mutation patterns by reducing the mutation intensity that is arbitrarily guided by the defender to weaken the avoidance itself, leading to a dynamic decoy bed, rather than being limited to unconditional avoidance of random attacks. In addition, this LPC-MTD mutation selection strategy can be realized based on the organizational mutation threshold as well as deep combination with organizational OSINT-based HS-Decoys, depending on the avoidance strength, attack surface exposure, and attenuation of the MTD mutation function. It will also be possible to advance defender deception for organizational MTD selection, calculate trade-offs, consider the OSINT influence on the network separation environment, and standardize OSINT-based resilience and agility.

Thus, the compound equation of LPC-MTD was configured, as per Equations (1) and (2) in Table 2 based on perturbation and Bellman sampling because of the principles of MTD [5,9,47–52].

**Table 2.** Definition of MTD principles and major equations for configuration of LPC-MTD.

| Principles of LPC-MTD | Description of Related MTD Equations |
|---|---|
| OSINT-based perturbation for artificial leakage (P) | Perform artificial disinformation to cause misunderstanding under attackers. $P = \Pr[L = l \mid S = \mu(s)]$,   (1) |
| Value iteration-based batch sampling (VS) | Compute statistical mutation batch pool based on Bellman iteration sampling, which is more advanced than random or proportional sampling. $VS^n(i) = \min\limits_{\tau_i, m_i \in M}\left[c_{i,j} + \sum \; \widetilde{m}_{i,j} VS^{n-1}(j)\right]$, (2) |

Algorithms 1 and 2 shows the pseudocode of the LPC-MTD equations in Table 2, and it is normalized with reference to the association and combination between HS-Decoys, and related mutation variables that will be updated.

---

**Algorithm 1.** Value iteration-based batch sampling in LPC-MTD.

---

**Input**: T, $\overline{\tau}$, $\underline{\tau}$, C, $\theta$        **Output**: M*
$\forall x \in T$, t = 0, $VS^0(x) = 0$;
**while** $C\underline{VS} > \overline{VS} - \underline{VS}$:
    t = t + 1;
    **for** $x \in T$:
    $v = \infty$;
    **for** $\tau = \underline{t}$; $\tau \leq \overline{\tau}$; $\tau = \tau + \theta$:
        $M' = \underset{M}{\mathrm{argmin}}\, VS^t(x,\, M,\, \tau)$, $v' = VS^t(x,\, M',\, \tau)$;
        **if** $v' < v$:
            $M_x^* = M'$, $\tau_x^* = \tau$, $v = v'$;
    $VS^t(x) = v$;
    $\overline{VS} = \underset{x \in T}{\max}\left|VS^t(x) - VS^{t-1}(x)\right|$;
    $\underline{VS} = \underset{x \in T}{\min}\left|VS^t(x) - VS^{t-1}(x)\right|$;
**return** M* = $\overline{VS} - \underline{VS}$;

---

**Algorithm 2.** OSINT-based perturbation for artificial disclosure.

---

**Input**: $\overline{l}$, $\underline{l}$, s **Output**: P*
t = 0, listavg = [], $\mu(s) = \Pr[S = s]$;
**for** $l = \underline{l}$; $l \leq \overline{l}$; $l = l + \theta$:
    t = t + 1;
    $\overline{P} = \underset{s}{\max}\left|\mu(s)\right| \sum\limits_{l} P(l\,|\,s) LC$;
    $\underline{P} = \underset{s}{\min}\left|\mu(s)\right| \sum\limits_{l} P(l\,|\,s) LC$;
    **if** $\overline{P} - \underline{P} < LC\underline{P}$:
        listavg(t) = $\overline{P} - \underline{P}$;
    **else**:
        **continue**;
**return** $P* = max(listavg)$;

---

### 3.1.3. HS-Decoy

A decoy could either indicate a cognitive skewness-inducing deception component that encourages malicious internal and external attacks on a false target or a dynamic sandbox for blocking chain attacks. It is the complex multi-tenant concept that is composed of a large amount of false data, and it appears to be valuable to an attacker because it is similar to normal services or important information and, therefore, is highly enticing [53].

A decoy should allow legitimate users to distinguish it from actual information and operations, and attackers should not be able to distinguish it without prior knowledge of the target system. Further, it can be used for persistent attacker inferiority triggering based on access event count, intrusion tolerance and restoration, and attacker initial cognitive skewness induction and isolation [54].

In summary, decoys have complex detailed properties, such as functional reliability and discrimination for deception, initial attack triggering and skewness possibility according to the defender's intention, constant exposure to trigger infringement attempts, dynamic diversity for induction, prevention of redundancy between other decoys and non-interference for abstraction, distinguishing between legitimate users and attackers, immediate early detection of invading attackers, and access control that is based on decoy-specific deceptive roles.

The compound equation of OSINT-based HS-Decoy is shown in Figure 3 and it is based on Equations (3)–(9) in Table 3, with a focus on the related decoy principles [54–57].

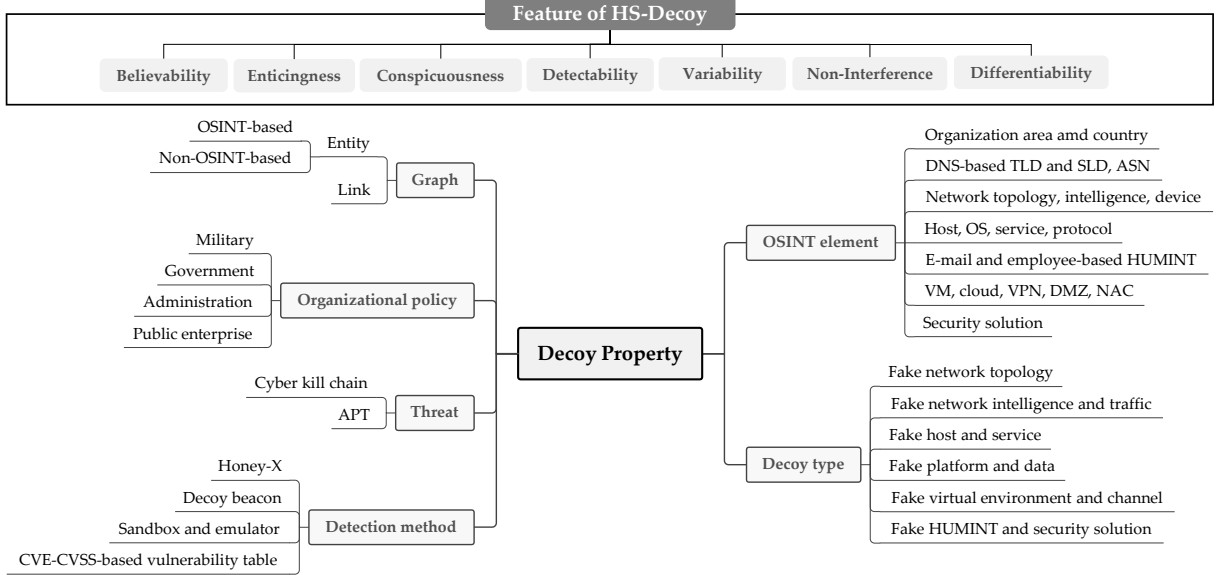

**Figure 3.** Lower-level overview of HS-Decoy with major features.

**Table 3.** Configuration of organizational OSINT-based HS-Decoy properties and major equations.

| Properties | Description of Related Decoy Equations |
|---|---|
| Believability (B) | Minimize the distinction between protected objects and decoys. $Pr\left[Exp_{A,H,O}^{believe} = 1\right] = \frac{1}{2}$, (3) |
| Enticingness (E) | Cause intentional compromise by attacker preferences. $Pr[o \rightarrow O \| o \in P] = Pr[h \rightarrow O \| h \in H]$, (4) |
| Conspicuousness (C) | Continuously exposes the attack and exploration surface artificially to the attacker to the defender's advantage. $\prod_{i=0}^{n} Pr[V_i] > \delta$, (5) |
| Detectability (DE) | Detect intrusive behavior of an attacker in decoys. $Pr[h \rightarrow O : CD_{A,h} = 1] \geq \epsilon$, (6) |
| Variability (V) | Yield mutation traits and related decoys with organizational adaptability, performance, and persistence. $Pr\left[h \rightarrow H : Exp_{A,H,O,h'}^{believe} = 1\right] = \frac{1}{2}$, (7) |
| Non-Interference (NI) | Responsible for detecting and preventing exceptions for legitimate users who refer to the decoys. $Pr[CT_{D,o,h} = 1] = Pr[CT_{D,o,h} = 1 \| H]$, (8) |
| Differentiability (DI) | Allow legitimate users to distinguish between target and lure, but make it indistinguishable to attackers. $Pr\left[Exp_{D,H,O}^{believe} = 1\right] = 1$, (9) |

Based on these properties, the concept of a hierarchical social engineering decoy (HS-Decoy) that is based on OSINT for each governmental organization disclosed on a third party basis can be standardized. HS-Decoy, along with LPC-MTD, is a sandbox container for standardizing artefacts that are specialized for each attacker targeting a specific organization. Because of the artificial active attack contact point composed of false organizational dummy information with OSINT, it can improve the attacker's likelihood of attempting an attack, and make cognitive skewness in extreme favor of the defender. In other words, the key is to sustain the attacker's chain attacks for as long as possible in the decoy isolation environment by minimizing the attacker's suspicion. This makes it possible to apply a disinformation-based vulnerability advertisement and intentional infringement allowance scheme according to LPC-MTD's loose attack avoidance in the upper layer, while applying HS-Decoy as a detailed deceptive environment for enticing, inducing, and isolating attackers in the lower layer. In the proposed model, hierarchical organizational OSINT groups were used to construct a decoy bed that was based on the basic properties of decoys [55–57]. Subsequently, to reflect real-time change in OSINT trends, reputations of third parties, LPC-MTD mutation cycle selection weight, and gradient in the HS-Decoy-based decoy bed, we multiplexed them on an independent distributed multi-tenant basis. We also considered a scheme for enforcing the defender superiority in an asymmetric attack–defense relationship by defining connectivity, dependence, and ripple power within HS-Decoy as a graph because of the entropy correlation of each OSINT.

### 3.2. Design Principles of Proposed Deception Model

3.2.1. IODA, Integrated OSINT-Based Deceptive Architecture

We present the integrated OSINT-based deceptive architecture (IODA) based on SDN using multiple deceptions, like Honey-X, embedded decoy, and social engineering tool, in this section in order to apply and verify LPC-MTD and HS-Decoy. The LPC-MTD concept was introduced in the upper layer with a focus on the MTD function based on the DRHM [43] strategy, while the OSINT-based HS-Decoy was multiplexed into a single kernel-based distributed sandbox container in the lower layer. Along with LPC-MTD based multi-layered induction, isolation, and monitoring, it was conceptualized, as shown in Figure 4, to apply the activeness and adaptability in decision making of HS-Decoy based on DDD [57] strategy. Next, we describe each component of the SDN-based IODA.

1.  Integrated management module for LPC-MTD mutation (IMM): this is a central management module that controls and manages all of the sub-modules in IODA using periodic control packet-based identification, authentication, authorization for each module, and comprehensively performs OSINT-based mutation. The module collectively performs the following: LPC-MTD-based structural mutation for docker-based sandbox, such as a legitimate server pool and deceptive server pool, TDM-based attacker detection and monitoring, artificial disinformation projection through OSINT-based HS-Decoy, deception of attack surface form, and migration-based induction.

    –   Interactions between IMM and LSM, DSF: in order to utilize all of the LPC-MTD concepts to project and expose organizational OSINT-based HS-Decoy-related disinformation, there is a process to distinguish an illegitimate user using IMM. This will attract the illegitimate user to DSF, and legitimate users will be routed to LSM. An illegitimate user could either be a legitimate user who is abusing their access or a malicious attacker, so the classification is supplemented based on the flow and rule table that were already configured in the deceptive server, and the update of the corresponding tables is determined based on the periodicity of proxy channels between IMMs and OSINT characteristics.

    –   Regarding the adaptive management of HS-Decoy in DSM and HMM, TDM is used as a trait extraction and processing module for attack detection and fingerprint scrubbing in IMM to calculate the effective information group.

    − Interaction between IMM and OOM: unlike other sub-modules, OOM is directly managed by a supervisor connected as proxy, and not by IMM. This secures independence for OSINT and performing access control-based encapsulation.

2. Legitimate server module (LSM): in the LPC-MTD concept, this is a legitimate server pool with limited mapping based on a flow table or a vulnerability table, and only users who request a network and host-based OSINT information group can use it legally at an arbitrary time in the organization for a predetermined length of time. If the detected user is a legitimate user, then the module provides organizational services within IODA and performs OSINT-based connection migration between legitimate users according to the LPC-MTD mutation, while performing the process of specifying and managing induction and isolation to deceptive servers within a specific DSF group based on network control packets in the case of attackers.

    − Interaction between LSM and DSF: independent mapping is performed for each LSM based on OSINT after performing both the creation and update of deception servers in DSF by constructing all of the flow and rule tables of LSM according to the conversion rules and mutation periodicity defined in IMM and DSM.

3. Adaptive deception server farm module-based HS-Decoy (DSF): this is an adaptive deception server that is involved in selective evasion, induction, isolation, and common vulnerabilities and exposures (CVE)-based non-cooperative simulation through LPC-MTD, and it induces and isolates illegitimate users or attackers into sandboxes. It operates based on the LPC-MTD-based decision strategy and HS-Decoy properties. It standardizes the sandbox by applying all of the available OSINTs after minimizing the discrimination between LSMs in the initial reconnaissance of the attacker and determining the similarity weight to ensure consistency for deception.

    − Interaction between DSF, DSM, HMM: when managing the group of DSF deception servers under the control of the IMM, the decoy management process in HMM improves the management efficiency of the HS-Decoy environment while complying with both the DSM's conversion rules and mutation periodicity.

    − Interaction between DSF and OOM: the exposed OOM is adaptively updated based on the main CVE simulation to express and de-inform the artificial attack contact points related to sophisticated attackers in a deception server belonging to the DSF group according to the supervisor's intention.

4. Deception server module controller (DSM): this is a deception active controller module that controls multiple DSFs mapped to a specific LSM, tracks and manages the current state of sandboxes and emulators within the DSF, and it monitors the current inferiority of attackers against defenders. It constructs conversion rules according to the attack-defense strategy and mutation periodicity established in IMM and OOM and performs modification and deletion of containers focusing on a single kernel of a docker-based server pool or farm corresponding to DSF, and additional mapping.

5. Automatic HS-Decoy Management Module based on OSINT (HMM): this is a decoy strategy management module that creates, distributes, and updates HS-Decoy, as well as the direction of the OSINT information group and CVE vulnerability for deception servers belonging to DSF. The decoy management process operates around the main attributes of the HS-Decoy. After selecting organizational OSINT in OOM according to the LPC-MTD mutation strategy, it passes environmental parameters to the DSF to characterize the HS-Decoy and related sandboxes by organizational traits.

6. Threat detection and monitoring module (TDM): this is a simulation module for extracting, processing, and continuously tracking features, such as social engineering attacker type or related attack patterns. This main module standardizes intelligence for each attack-defense scenario, and it specifies and manages all of the reverse connections, drive-by-download-based internal threats, information gathering and remote code execution-based external threats, deliberate or misuse-based insider threats, automatic scanning, and custom shellcode-based human and machine factors

with rule tables between IMMs. In addition, as a policy to identify random deception patterns, and to interfere with bypass attacks that attempt reverse disturbance, it performs fingerprint jamming separately, which is meaningful for the analysis of defender traits, mass production of false traffic, and insertion and modification of arbitrary dummy headers based on proxy between DSFs.

7. Organizational OSINT graphDB module (OOM): this is a social engineering OSINT information group management module based on manual enumeration and hierarchical DNS, which collects and standardizes public archive data. This module reinforces the traits and properties of HS-Decoy to configure multiple HS-Decoys in a distributed multi-tenant structure in detail into node- and edge-based information graphs, and it performs OSINT updates based on parallel pipelines to reflect organizational OSINT fluctuations. It also creates, distributes, and updates random OSINT based on organizational traits with HS-Decoy, and finally reflects the update gradient for each node by determining the reputation based on third party reliability and volatility to create attack-defense inferior relationships. Figure A2, in Appendix A, shows a detailed conceptual diagram that is related to the sub-modules, components, and processes in the OOM. The OOM management controller manages the detailed sub-modules, and communication for collecting organizational OSINT and HUMINT groups. The analysis module is used to process the collected OSINT information groups and calculate and normalize the weights for each property based on the calculation module. Subsequently, a decoy extraction module is used to distribute and de-inform the normalized OSINT information group based on HS-Decoy, which is finally standardized as a knowledge-based graphDB.

8. Sandbox or emulator: to disrupt the random attacker attempting to invade the OSINT-based deception server belonging to the DSF, this virtual execution container contains a statistically vulnerable network and host-based OSINT, such as IP, port, and service in the form of a rule table in order to artificially expose the initial attack contact points with HS-Decoy and CVE under the defender's intention. This container contributes to the creation of false fingerprints along with HMM in the presence of IMM to maximize the attacker's cognitive skewness and prevent bypass.

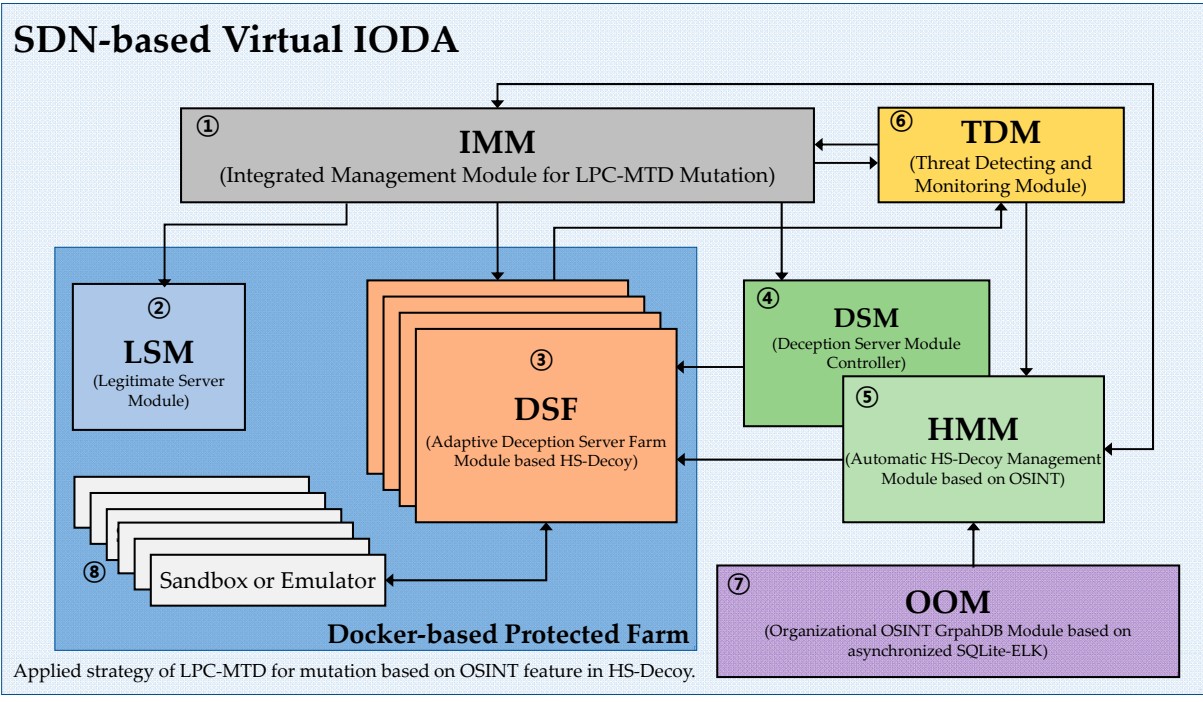

**Figure 4.** Overview of software defined network (SDN)-based integrated OSINT-based deceptive architecture (IODA) structure and related deceptive modules with LPC-MTD and HS-Decoy.

### 3.2.2. Detailed Module-Specific Decision Process for LPC-MTD and HS-Decoy

The multi-layered decision process is configured in IODA, as shown in Figure 4.
- Primary decision process: attack avoidance and request approval

(1)  When receiving a request for a network and host-based OSINT that was not previously promised in the LPC-MTD cycle or intentionally projected for disinformation, IMM immediately directs the requestor to the deception server in the DSF and then isolates it. When the requestor is identified as an illegitimate user or attacker, defender dominant behavior is executed, including cognitive skewness, minimization of suspicion, continuation of attacks in the isolation space, and backtracking with the HS-Decoy-based sandbox, and the behavior is monitored with TDM. The change is immediately reflected in the applicational deception concepts and modules to secure an adaptive update according to external entropy changes. Until the administrator's aperiodic intervention, this active containment is realized by performing a gradual repetition of this step for each attacker with LPC-MTD's periodic mutation as default.

(2)  When an arbitrary user requests a pre-promised legitimate OSINT information group, seamless connection-based migration is performed to connect to the legitimate server in the LSM. Subsequently, after identifying the requestor as a potential legitimate user or normal user, it reflects entropy change in the policy and rule table, and configures the use of the service provided by the legitimate server. Continuous monitoring is performed at this time through the detailed modules, such as IMM, TDM, and OOM, to detect any malicious behavior and determine whether the user was altered or malicious from the beginning.

(3)  The OSINT based on the exposure of authorized or unauthorized information is dynamically configured to minimize the attacker's suspicion and secure adaptability to external entropy changes from third parties, depending on the spatiotemporal mutation points and viewpoint or aperiodic LPC-MTD. At this time, the randomness of the mutation is created by a probability distribution with Bellman sampling

- Secondary decision process: attack detection, inducement, and isolation.

(1)  A legitimate server in the LSM is vulnerable to possible attack attempts and invasion from users who normally accesses the legitimate server and is a substantially malicious and prudent insider or a sophisticated external attacker.

(2)  When a primary legitimate user, who has passed the primary judgment, attempts malicious actions present in the policy table and vulnerability database, such as internal reconnaissance, theft, elevation of privilege, and remote code execution, the user is designated as an attacker, and it is immediately induced to the deception server and sandbox in the DSF, where active isolation is performed. Next, the attacker is monitored for repeated performance based on item (1) of the primary decision process. In other words, a previously assumed legitimate user is actually an attacker.

(3)  In addition, the lateral scanning of an attacker in the internal network located in a deception server farm is calculated as a deep reconnaissance activity for lateral movement and pivoting, and it is maintained based on cyber kill chain (CKC). Further, additional deception is performed by providing large amounts of false response traffic or disturbed fingerprints based on previously disinformed organizational OSINT.

The additionally applied vulnerability DB is constructed as a CVE table when configuring all of these organizational OSINT-based deception concepts, integrated architecture, major modules, cooperative processes, and decision strategies, as shown in Table 4. Each vulnerability was selected from the CVE lists with a focus on the ease of application and simulation implementation in the SDN-based HS-Decoy based on CAPEC [58] standard, and high CVSS focusing on Windows and Linux. Additionally, the attack contact points were formulated in detail based on port-based vulnerable services and protocols.

**Table 4.** CVE-based vulnerability table for disinformation and intended leakage with HS-Decoy.

| CVE ID | Vulnerability | Related Service and Protocol | CVSS 2.0 |
|---|---|---|---|
| CVE-2020-1350 | *SigRed*. Heap overflow, Remote code execution | DNS, TCP/UDP 53 | 10 |
| CVE-2020-0796 | *SMBGhost*. Remote code execution | SMBv3, TCP 139, 445 | 7.5 |
| CVE-2019-0708 | *BlueKeep*. Use-after-free, Remote code execution | RDP, TCP 3389 | 10 |
| CVE-2017-0144 | *EternalBlue*. Remote code execution | SMBv1, TCP 13,9 445 | 9.3 |
| CVE-2015-1635 | Remote code execution, Denial-of-service | HTTP, TCP 80, 443 | 10 |
| CVE-2019-6111 | Directory traversal | SSH, TCP 22 | 5.8 |
| CVE-2019-0232 | Remote code execution | HTTP, TCP 8080 | 9.3 |
| CVE-2018-15473 | Obtain information | SSH, TCP 22 | 5.0 |
| CVE-2017-7494 | *SambaCry*. Remote code execution | Samba-based SMB, TCP 139, 445 | 10 |
| CVE-2011-2523 | Remote shell, Backdoor | Vsftpd 2.3.4, TCP 6200 | 10 |

## 4. Experiments for Organizational Deception with LPC-MTD and HS-Decoy

In this section, the structure map and parameters of the proposed model with LPC-MTD and HS-Decoy are formalized based on normalization while using a true random generation-based Monte Carlo model and the flexible probability distribution that is based on the attack–defense scenario, and the comparative analysis with LPC-MTD and HS-Decoy deception experiments based on organizational traits is performed as a whole.

### 4.1. Definition of Experimental Metrics in the Testbed

In the experiments, to focus on LPC-MTD and HS-Decoy in IODA, OSINT-based attack-defense metrics were calculated based on the hierarchical tree from the perspectives of attackers and defenders, as shown in Figures A3 and A4 in Appendix A. Each tree was classified, in detail, based on the surfaces of the attacker and defender from the atomic point of view, the surface concept was subdivided into spatiotemporal performance, effort, cost, CKC, and APT, and an interface for variable estimation was also presented.

Figure A3 shows the metrics from the attacker perspective and, on the highest level, is categorized into the spatiotemporal OSINT specifications, and the effort and type of attacker. Further subdivisions were made for spatiotemporal intelligence, attack attempt, mutation cycle frequency, CKC, APT-based initial intrusion contact point, and attack chain continuity. These were then broken down into final lower attack-defense metrics. During the organizational OSINT-based experiment, the attack attempt, mutation cycle frequency, and CKC metrics were calculated as manipulation variables for producing the results for LPC-MTD and HS-Decoy. Figure A4 shows the metrics from the defender's perspective and, on the highest level, is categorized into organizational communication traits, LPC-MTD, and HS-Decoy. These are further subdivided into performance and overhead factors in the environment that simulate each organizational trait, LPC-MTD mutation factor, and OSINT and decoy property for the HS-Decoy component. Again, these are broken down into the final lower metrics. The attack surface from the attacker's perspective in the presented tree is the sum of multidimensional attack vectors that represent the potential attack paths, in which the exploitation of random vulnerabilities is available, while the exploration surface is the codomain of the attack surface acquired by a random attacker when performing various pre-attacks, such as reconnaissance, weaponization, and exploitation. Thus, the concepts will be used to verify and update the deception efficiency of LPC-MTD and HS-Decoy in relation to the performance, effort, and overhead of attackers and defenders in the attack-defense scenario. In addition, the possibility of maximizing cognitive skewness and minimizing suspicion, the possibility of a false attack for the attacker's deception, and the possibility of substantive simulation of the exploration surface can be additionally presented. For this experiment, the deception results were calculated based on the metrics in Figures A3 and A4. Further metrics include environmentally dependent variables in the test bed, and single or multiple attack-defense metrics that were the basis of analysis. In addition, in order to analyze the deception relevance and sequence based on OSINT, the

impracticality of SDN was attenuated as much as possible by adding dynamic weights to the execution time and scenarios for each metric.

### 4.2. Configuration of Experimental IODA Testbeds and Deception Sequence

The test bed for the organizational OSINT-based deception experiment was configured in detail, as shown in Figure 5. Because it is possible to attenuate the skewness of results according to experimental change or dependency issues between variables when compared to legacy MTD according to manager intervention based on a fixed virtual environment, the configuration of the IODA and the CVE-based attack-defense simulation with OpenFlow-based SDN was chosen to control the loose mutation period of LPC-MTD in the deceptive environment to which organizational traits were applied with the SDN controller. The creation, distribution, update, and normalization of organizational OSINT-based HS-Decoy was also easier to implement within SDN.

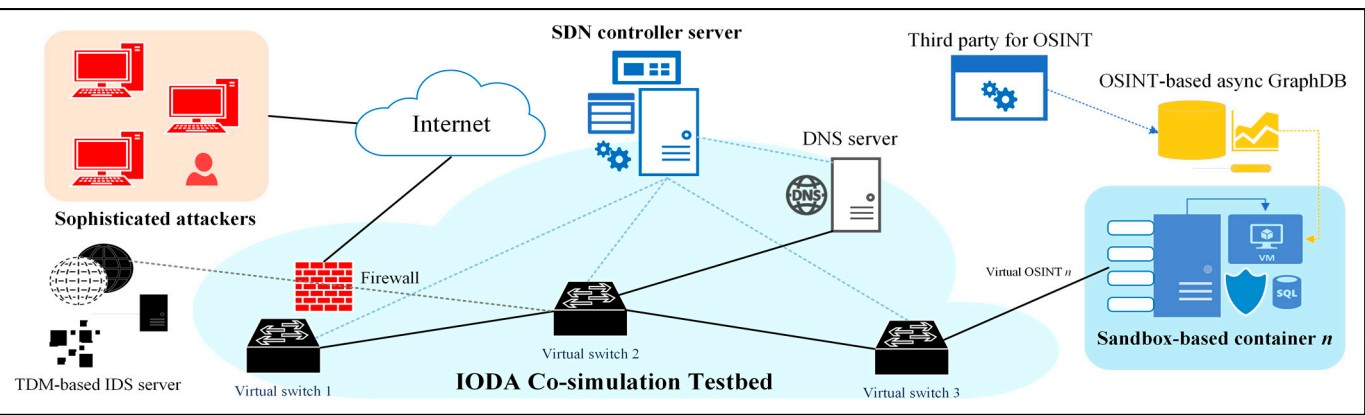

**Figure 5.** Overview of SDN-based IODA testbed with deceptive concepts, i.e., LPC-MTD and HS-Decoy.

In addition, as a target for direct mutation of LPC-MTD in the test bed, direct mutation of the authorized network band group used for normal service supply may not be suitable for ongoing organizational service delivery and migration goals. Therefore, a virtual network communication channel was additionally expanded and defined as a mutation target of LPC-MTD based on the network fingerprint range in the OSINT-based HS-Decoy, which was previously promised in the random mutation time interval. In addition, the flow and rule tables in the SDN controller were configured to communicate with each other in a dynamic pair structure between the real communication channel and the virtual communication channel, and the third party-based mutual update flow between the TDM-based IDS server and the organizational OSINT-based graphDB was calculated, and deceptive adaptability in IODA according to external entropy changes was secured.

Accordingly, the overall experimental variables that were related to the testbed in which the concepts, OSINT, LPC-MTD, and HS-Decoy were cultivated, are shown in Tables A2–A5, and deceptive sequences are shown in Figure 6, and they should be considered together.

The deception sequence corresponding to Figure 6a is a reactive sequence that is related to MTD mutation and decoy deception in the SDN where a malicious attacker requests access to a legitimate server in IODA and LPC-MTD and HS-Decoy based on network information, such as IP, port, and domain. If the network specifications in packet requested by a random malicious attacker are not permitted at any point, but trying to induce and isolate the attacker to an isolated sandbox in IODA with the defender's dominant intention for attack analysis without robustly evading the attacker in advance, the network-based OSINT group corresponding to the information group that is requested by the attacker is artificially created, inserted into the fake response packet, and then delivered with the intention of minimizing the attacker's suspicion and performing only

inferior attacks. Subsequently, the attacker is induced and isolated to the deception server farm and related internal networks in IODA that are falsely configured based on the OSINT information group configured by the defender. Figure 6b is similar, but it shows a scenario where it is easy to negatively spread the attack chain to the platform and host units, such as with file protocols, including SSH and FTP, SMB, and NetBIOS for message sharing, and RDP for remote. The difference is performing LPC-MTD and HS-Decoy with platform and host-based OSINT groups.

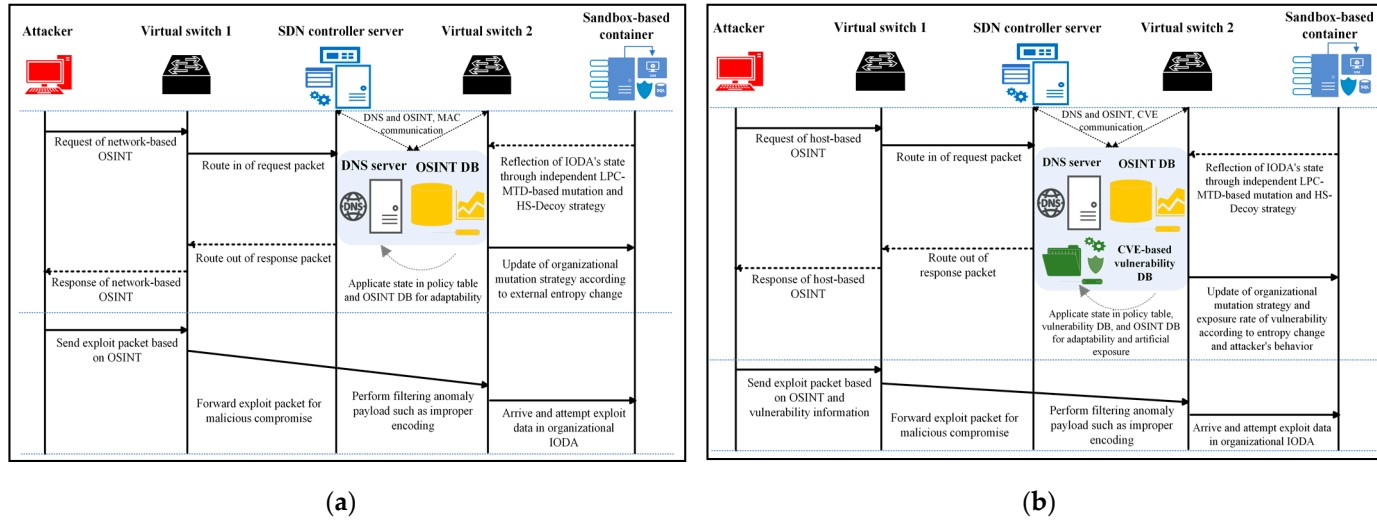

(**a**)                                                                                    (**b**)

**Figure 6.** Examples of deceptive sequences with LPC-MTD in HS-Decoy-based sandboxes. (**a**) Mutation sequence of network-based OSINT elements, (**b**) mutation sequence of platform and host-based OSINT elements.

### 4.3. Experimental Results and Related Organizational Sensitivity Analysis

By applying the organizational OSINT and HUMINT traits in the SDN-based IODA experimental environment, as shown in Figure 5, we analyzed and compared LPC-MTD and HS-Decoy with existing deception elements in terms of attack success probability, artificial information exposure ratio, deceptive probability, and enticingness probability, as shown in Figure 7. Figure 8 shows the sensitivity analysis for the LPC-MTD and HS-Decoy-based optimized model in the test bed, as well as the comparisons and analysis of the following variables by final normalization with true generation-based Monte Carlo: attacker and defender's defense efficiency by mutation time cycle; spatiotemporal attack cost by OSINT mutation range; port and service within OSINT; detailed mutation selection efficiency by protocol; defender superiority by MTD batch pool size based on Bellman sampling; in-packet attack payload or IODA control overhead by defender dummy fingerprint size; and, the attack execution time of random attacker according to the number of CVE vulnerabilities. Figure 9 shows the results for conceptually presenting the organizational optimal proactive avoidance time cycle for LPC-MTD and HS-Decoy in terms of operational security by deeply applying it to the traits of the US Armed Force, US Unified Combatant Command, and the US Intelligence Community among public OSINT groups applied in IODA and performing attack-defense simulations (the OSINT groups used do not have actual critical network, host, and human specifications). Figure 10 shows the results of the calculation of CKC-based attack-defense effectiveness and defender superiority according to the degree of deceptive surfaceization for each attacker and defender within IODA, focusing on multidimensional entropy-based standardization related to changes in the internal and external environment for each CKC scenario along with compromise likelihood visualized by reconnaissance, weaponization, and exploitation stages.

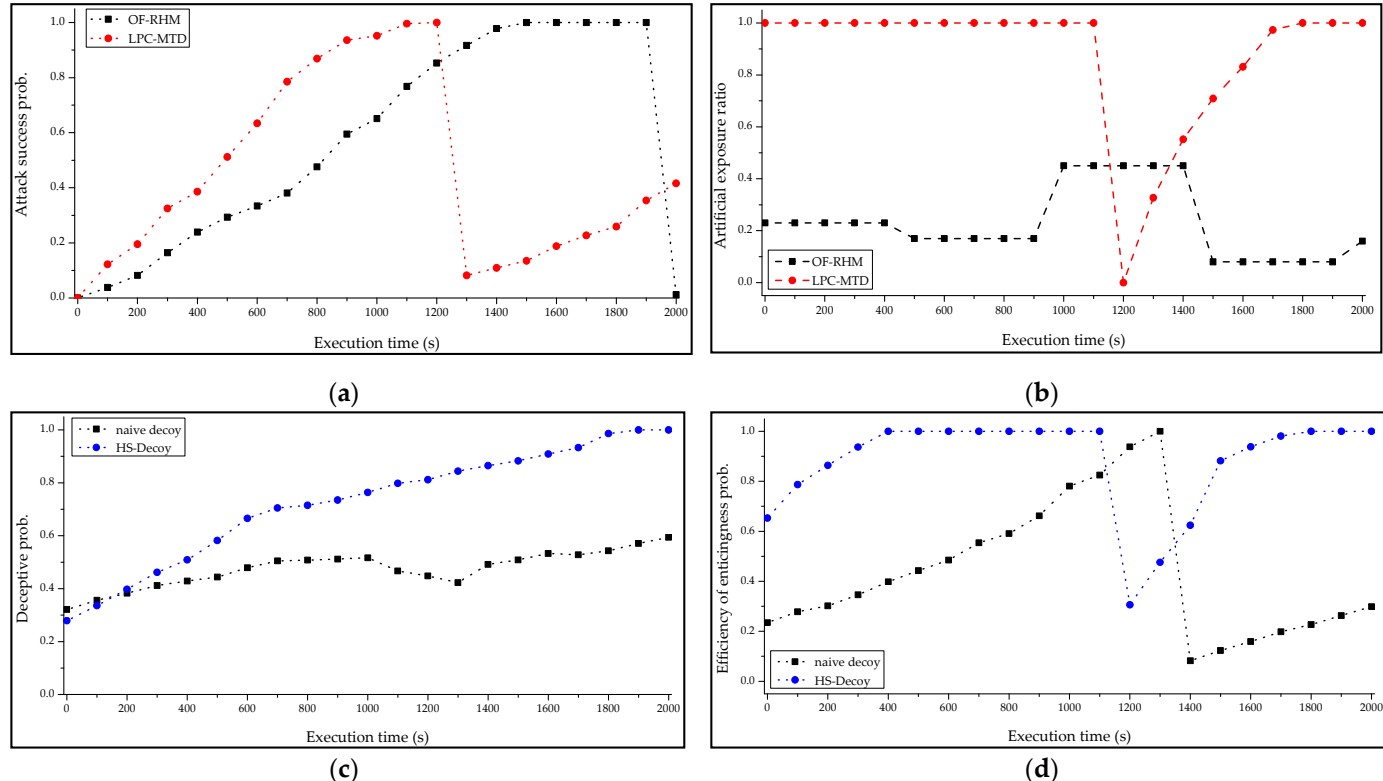

**Figure 7.** Comparison of results of the existing deception concept and the proposed models. (**a**) Attack success probability between OF-RHM [59]-based naïve MTD and LPC-MTD, (**b**) artificial organizational exposure ratio, (**c**) deceptive probability between naïve decoy and HS-Decoy, and (**d**) efficiency of enticingness probability.

The attack success probability that is shown in Figure 7a is a probabilistic comparison metric that atomically normalizes the spatiotemporal attack cost that is consumed before the attacker occupies the final compromise point within the organizational topology surfaced based on the CVE vulnerability or decoy beacon within the test bed where the LPC-MTD or existing OF-RHM was applied in a defender dominant manner. This is related to detectability and believability, as shown in Figure 3. It shows that the primary deception efficiency for progressive attacker isolation is twice as high, as attackers occupy the final compromise point earlier than with OF-RHM-based conventional MTD. This shows that when isolating the attacker in the defender's isolated sandbox, the LPC-MTD-based mutation does not simply perform unconditional evasion against the attack, but it artificially lowers the mutation strength under the criteria that guarantees the defender's superiority to lower the evasion strength, leading to the prompt initial occupation of the decoy environment with intentionally exposed false information. In addition, after the defender dominates the compromise point, the attacker reconfigures the attack chain according to the LPC-MTD shuffling, which is immediately re-performed under the defender's intervention. It can also be seen that the efficiency of the reconstructed attack chain decreased 2.8 times on a probabilistic gradient basis when compared to the previous one. This shows that LPC-MTD mutated based on Bellman-based sampling according to intrusion.

The artificial exposure ratio shown within Figure 7b is a comparative metric that conveys the degree to which IODA's attack and the traits of the exploration surface created based on the uniqueness of OSINT of a specific organization can cause attack attempts by attackers according to the attack-defense correlation. This is related to conspicuousness and variability, as shown in Figure 3. It can be seen that the LPC-MTD contains an intentional information disclosure ratio that is eight times higher on average than conventional OF-RHM. Even after identifying the primary initial compromise by the attacker, it can be seen that the exposure ratio that is based on OSINT traits of the same organization is also

improved 11 times higher based on the gradient, and it is quickly restored and reconstructed as a separate deceptive surface that is less related to the previous surface as a measure to ensure the gradual compromise behavior quickly in a defender dominant manner while limiting the effectiveness of the previous defender trait possessed by the attacker to an extreme degree. As in Figure 7a, this shows that the LPC-MTD, unlike the existing MTD, deliberately supplies the OSINT-based false information to entice the attacker while reactively attenuating the evasion intensity to isolate the enticed attacker. Through the ratio aspect that only increases from 1000 to 1400 s within the existing OF-RHM and maintains a 0.08 and 0.23 exposure ratio, it can be seen that it was raised by the attacker's sequential attacks, and it was not caused by artificial exposure and disinformation, as with LPC-MTD.

The deceptive probability enticingness shown in Figure 7c,d are probabilistic metrics in which HS-Decoy or existing decoy-based sandbox normalizes the secondary deception efficiency property to maintain the attack attempt under the defender's control after the initial attack (e.g., reconnaissance, pre-weaponization) and it implies the maximum isolation probability within the sandbox. It is directly related to all properties, focusing on the enticingness shown in Figure 3. Based on these metrics, it can be seen in (c) that HS-Decoy has a 3.8 times higher deception efficiency when compared to existing decoys, and the improvement of gradient-based deception quality is also positive. Figure 7d shows that HS-Decoy's enticing efficiency quickly regenerates and distributes new decoys that are hardly related to the previous decoys and they have a connection with the referenced OSINT traits after the initial occupation and can be maintained at a probability of 1.0, where existing decoys exhibit a probability gradient. These results show that the extremes of the attacker's cognitive skewness and the maximum persistence of the attack chain in the sandbox are due to the uniqueness of the correlated entity and link-based hierarchical OSINT through HS-Decoy-based disinformation. In this experiment, the cognitive skewness was based on the OSINT-based false fingerprint that was initially assigned. In Figure 7c, it is shown that the difference in the efficiency of deception between existing decoys increases linearly over the execution time. In addition, the efficiency improvement that is related to the creation and distribution of false information is also demonstrated in Figure 7d in the standard that guarantees the highest possible enticingness. The increase in gradient shows that OSINT-based HS-Decoy is quickly reorganized after initial occupation.

This shows that the efficiency of the OSINT-based LPC-MTD and HS-Decoy deception concept has improved on the existing OF-RHM-based MTD and naïve decoy. Next, we discuss the comparisons and analysis that are presented in Figure 8, a set of sensitivity analysis results for IODA optimized based on organizational OSINT traits.

Figure 8a shows the attack–defense efficiency related to the effectiveness of the defender's traits limited by the time slot length that determines the LPC-MTD mutation cycle in IODA. This shows that, when calculating the LPC-MTD mutation cycle, if the relative timestamp-based time slot is between 1 and 50 s, the highest spatiotemporal attack cost and effort can be given to the attacker in a defender dominant manner. However, the attack cost and effort of the attacker decreased by 158% when compared to the previous, as the defender's dominant relationship is greatly reversed when the time the range is expanded to 200 s. In this case, although frequent LPC-MTD mutations impose a major overhead factor in the defender and SDN controller central control groups, it also shows that a defender's sandbox that is subject to frequent final compromise by an attacker can impose an even higher overhead than loosening the mutation cycle to reduce the performance overhead. Because the effort-based attacker's superiority is improved by 10% in the 250 s range as compared to the previous, showing that in the final aspect (a), the minimum time interval for maintaining the defender's superiority from the perspective of proactive avoidance within the SDN-based IODA should be configured to less than 170–180 s. If the concept of robust avoidance is not considered, then there will be a few security issues that are related to practical deception defense, as the isolated sandbox exists, even if it is assigned a longer time interval period. However, the primary initial cognitive skewness in LPC-MTD, which can entice an attacker based on MTD, will not be performed significantly.

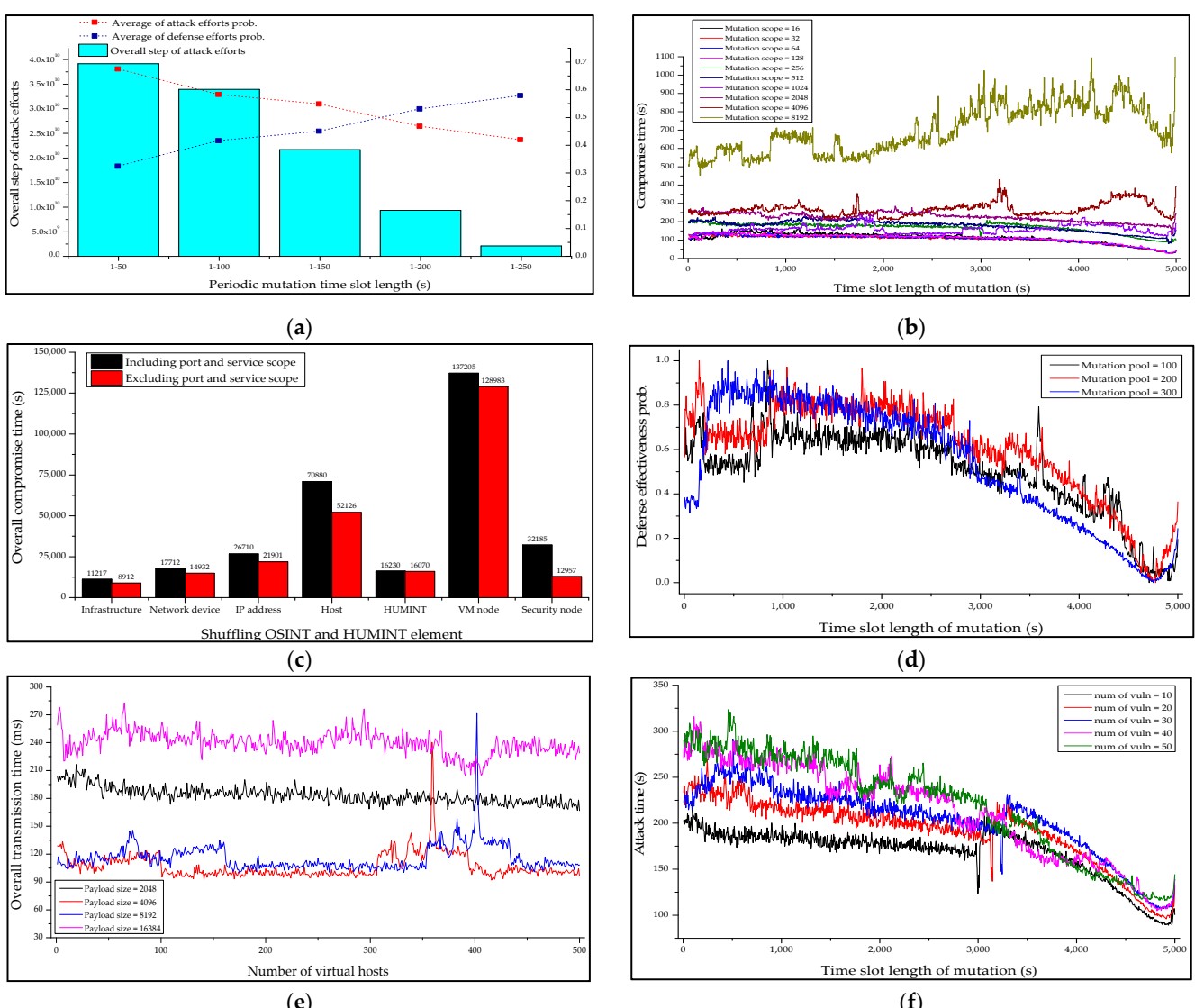

**Figure 8.** Comparative sensitivity analysis of organizational IODA with optimization of LPC-MTD and HS-Decoy. (**a**) Periodic mutation time slot, (**b**) mutation scope of OSINT and HUMINT, (**c**) mutation port and service scope of OSINT, (**d**) mutation selective batch pool size, (**e**) exploit or dummy packet size in SDN, and (**f**) number of CVE vulnerability.

Figure 8b shows the spatiotemporal attack efficiency of an attacker that is related to the calculation of the mutation range for OSINT and HUMINT in HS-Decoy in LPC-MTD to improve the disinformation efficiency of HS-Decoy. This shows that, when calculating the HS-Decoy mutation range, the effectiveness of the defender's attack and exploration surface improves linearly until the candidate area is defined as 4096 or less, and the defender's dominant relationship improves exponentially from 8192 and higher. It also shows that, as the uncertainty of the reconstructed HS-Decoy increased along with the mutation range, the attacker's inferiority increased, but the defender-control-based deception stability decreased. Finally, to improve the defender's superiority of OSINT and HUMINT-based HS-Decoy as a single parameter, the mutation range should be extended to 2048 or more in order to provide the concept of proactive avoidance and high randomness. Conversely, when controlling the reconstruction of HS-Decoy under the intervention of the defender, it should be limited to 1024 or less.

Although similar to Figure 8b in terms of direction, Figure 8c shows the improvement in effectiveness of the defender related to whether to calculate the mutation for each element formed after weighting is added, focusing on port, protocol, and service elements

among OSINTs in HS-Decoy in order to improve virtual mapping efficiency based on the CVE vulnerabilities presented in Table 4. This shows that, when estimating the mutation by socket port and protocol, the service-based variation element in LPC-MTD's HS-Decoy, the security node, and the host produced the highest defender superiority among the hierarchical elements in HS-Decoy. On the contrary, HUMINT based on human data, such as employees and e-mail, had no correlation. It also shows that all other elements were improved, except for HUMINT. Finally, Figure 8c shows that, when performing CVE-based attack-defense simulations using HS-Decoy, configuring separate sub-relationships as detailed elements, such as ports, protocols, and services in most OSINT classifications, except HUMINT, could degrade the attacker's inferiority under the defender's intention. In this case, when seamless connections and services are not considered, there will be a few availability issues that are related to the detailed elements. However, the purpose of minimizing the seamless problem caused by MTD is also included in the purpose of introducing LPC-MTD, so the performance overhead analysis that is related to the calculation of the corresponding mutation range in IODA should be preempted.

Figure 8d shows the effectiveness of the defender's superiority related to the selective MTD mutation by pool after giving the MTD batch by configuring the Bellman-based batch sampling secondarily, rather than limiting the random mutation within the calculated range after pre-calculating the concept of scope to improve the efficiency of de-information of HS-Decoy as the defender efficiency enhancement that is suggested in Figure 8b. This shows that, when the HS-Decoy mutation range of LPC-MTD is stratified and selectivity is added, and the batch pool size was set to 100, which was based on not exceeding 5% of the maximum allowable delay threshold within the SDN, the lowest decoy mutation uncertainty and defense efficiency were achieved, on average. Conversely, when as the batch pool size was set to 200, the highest decoy mutation uncertainty and defense efficiency were achieved, on average. A batch pool size of 300 or more without considering the delay resulted in overhead due to frequent mutation cycle reset up to 200 s. Finally, if the HS-Decoy mutation stratification of LPC-MTD was performed with reference to (d) and the defense efficiency was improved with a single parameter by presenting randomness, related sampling-based parameters should be limited to less than 100 when the sampling was extended to a value between 100 and 200, while minimizing uncertainty.

Figure 8e shows the attacker's payload that is related to the packet control issue in the SDN controller or transmission control overhead by the defender's dummy fingerprint size among potential overheads in a deception sequence, as shown in Figure 6. When customizing the maximum allowed packet size as the maximum segment size (MSS) according to the standard in which the protocol traits were fixed to TCP without exceeding the delay threshold within the SDN, an average of 118 ms was derived in the time span from 300 s to 450 s, except for the rapid increase in overhead due to rule table exception processing in the SDN controller when defining the payload size (or dummy fingerprint size) from 4096 bytes to 8192 bytes. The sudden change in overhead for payloads of 2048 and 16,384 bytes was low (the sudden change for 16,384 bytes is higher than that of 2048 bytes with a gradient), being recorded at 187 ms and 258 ms, respectively. This aspect was calculated while performing the LPC-MTD mutation policy, as it is based on the rule table in the SDN controller, regardless of the byte multiple difference. Finally, when the transmission control overhead in IODA was attenuated to a packet size with reference to (e), it should be limited to a range between 8192 and 2048 bytes. Conversely, when the sudden change in overhead is minimized, it should be calculated as 2048 bytes.

Figure 8f shows the scale of attack effort, depending on the number of CVE vulnerabilities used as final compromise points in the defender's sandbox. This shows that, as the number of vulnerabilities increased, the attack time increased linearly by 10%.

Next, after enhancing the organizational infrastructure that was replicated within IODA based on public OSINT traits related to US Armed Forces, US Unified Combatant Command, and US Intelligence Agencies, in Figure 9 we present a result set that calculated

the operational security LPC-MTD and HS-Decoy mutations that were optimized for each organization by performing attack–defense simulation within a limited range.

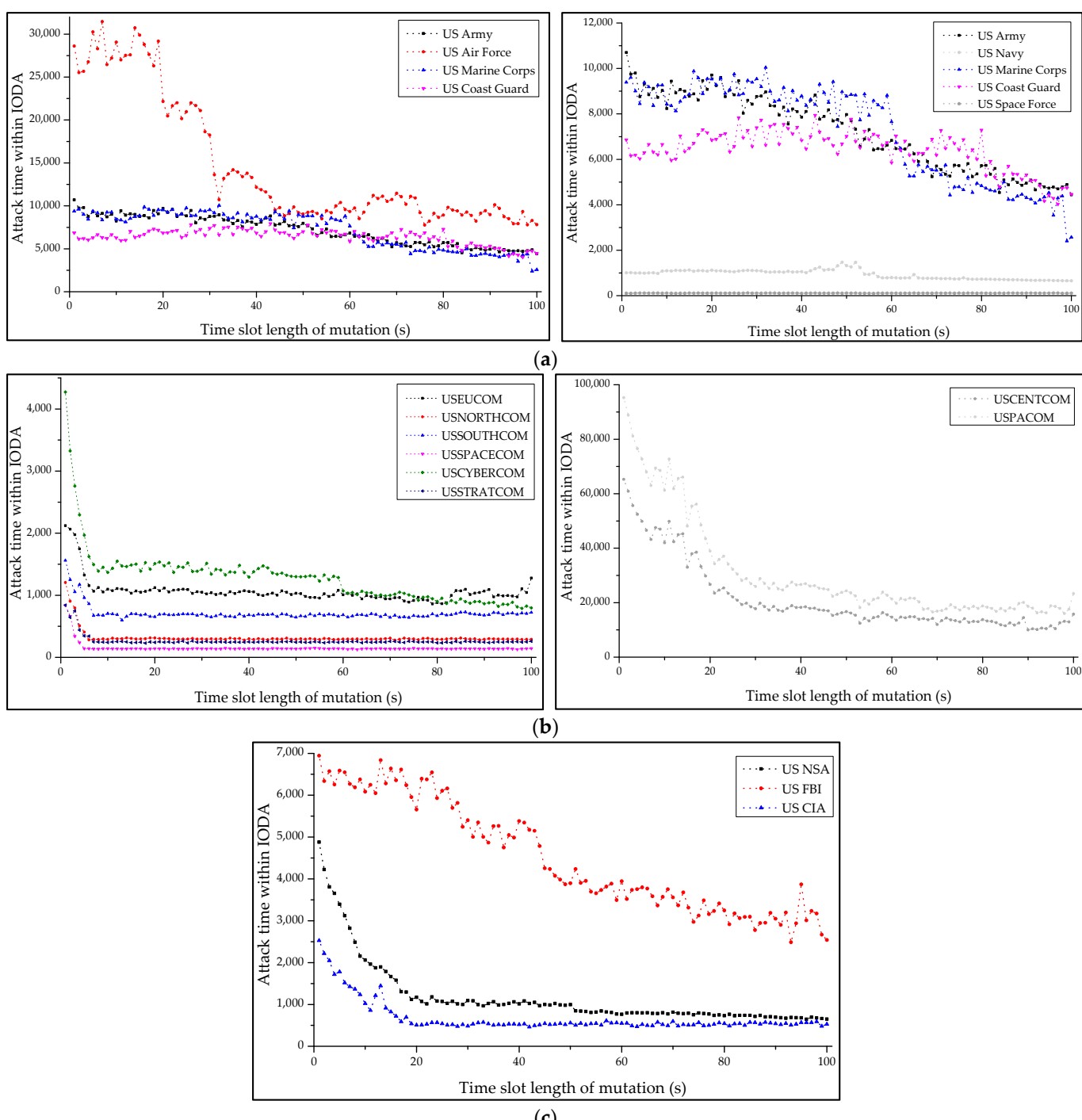

**Figure 9.** Comparative results of public OSINT-based environments by US organizations. (**a**) US Armed Forces-based, (**b**) US Unified Combatant Command-based, and (**c**) US Intelligence Community-based.

Figure 9a conceptually shows the attacker's inferiority based on time slot length optimized in relation to the US Military-based deception security environment by applying public OSINT information groups based on active military specifications that are related to US Army, US Air Force, US Navy, US Marine Corps, US Coast Guard, and US Space Force among eight detailed military organizations, including the Department of Defense,

which make up the US Military. First, it is proved that the US Air Force produced the highest prophylactic deception efficiency and agility on average when compared to other organizations and obtained the highest defender superiority when the LPC-MTD mutation cycle was 7 s. It also shows that the most optimized trade-off was produced at 19 s based on a downward gradient when aiming for a high overhead reduction according to frequent mutations in the US Air Force-based environment while maintaining high proactive security. The US Army, US Marine Corps, and US Coast Guard show similar tendencies, but the US Army produced the lowest overhead for changes in the organizational environment due to mutation, while accepting attacker compromise most reliably, on average, as the mutation time slot length increased when compared to other organizations. On the other hand, the US Coast Guard produced low security changes and high precautions on average, even with changes in time slot length. However, it showed the highest overhead frequency according to the change in each mutation cycle. Although producing high deception efficiency on average, the US Marine Corps showed higher uncertainty in the performance overhead processing according to mutation as compared to the US Army. Thus, the optimized trade-off based on the LPC-MTD mutation cycle for each of the three organizations showing similar graphs proved to be 23 s for the US Army, 25 s for the US Marine Corps, and 21 s for the US Coast Guard. Finally, the US Navy and US Space Forces produced the lowest degree of security change, preventiveness, and overhead when compared to all the other organizations. Further, the US Navy's trade-off was proven to be 26 s and the US Space Forces to be 2 s. Finally, Figure 9a theoretically presents optimized deception security within limited OSINT while considering the overhead in the US Military-based test bed.

Figure 9b shows the attacker's inferiority optimized by hierarchical command and control system specifications as USPACOM, USEUCOM, USSOUTHCOM, USCENTCOM, USNORTHCOM, USCYBERCOM, USSTRATCOM, and USSPACECOM, among the organizations constituting the US Unified Combat Command. Unlike the US Military, all of the command centers with the exception of USCENTCOM and USPACOM produced the highest predictiveness between 1 and 3 s, followed by a sharp downward gradient as the time interval increased. Additionally, the optimized trade-offs were derived in sequence between 5 and 9 s. This is because only a few OSINTs were applied to IODA and the deception sequences, limiting the simulation's influence range by weight assigned hierarchically. On the contrary, USCENTCOM and USPACOM showed a downward gradient that was related to a decrease in the defender's superiority according to a change in the mutation cycle because the number of applied hierarchical OSINTs were three to five times more than other organizations, and the trade-off was also calculated as 13 s. Finally, the optimized deception security could also be conceptualized while considering the overhead in the environment of the US Unified Combat Command, as shown in Figure 9b.

Figure 9c shows the attacker's inferiority optimized with the specifications of the information community related to the NSA, FBI, and CIA, among US Intelligence Agencies. Similar to the US Unified Combat Command, all of the intelligence agencies produced the highest predictive security at 1 s, and it also showed a gradual downward gradient in line with an increase in the time slot length. However, the FBI had 2.8 times the number of published OSINTs compared to the NSA and CIA, so the graph shows a higher sequential descent. The NSA and CIA produced a trade-off at 19 and 20 s, respectively, and the trade-off is composed of 28 s. Again, this can be attributed to only a few available public OSINTs, which limits the influence of each OSINT element. In addition, it is also related to the fixation of the mutation surface, which occurs because the OSINT information group contains fewer elements that improve the deception sequence efficiency, such as hosts and services, except for authorized IP or VPN. Finally, in Figure 9c, it is possible to conceptualize the deception applicability and security in the Intelligence Agency environments.

The trade-offs that are based on public OSINTs are summarized in Table A6 in Appendix A. Finally, we calculated the attack–defense efficiency and defender's superiority according to the deceptive surfaceization at the attack-defense point as the compromise

likelihood of the CKC phase, and present, in Figure 10, the result set of surfaceization performance based on the multidimensional transfer entropy gradient.

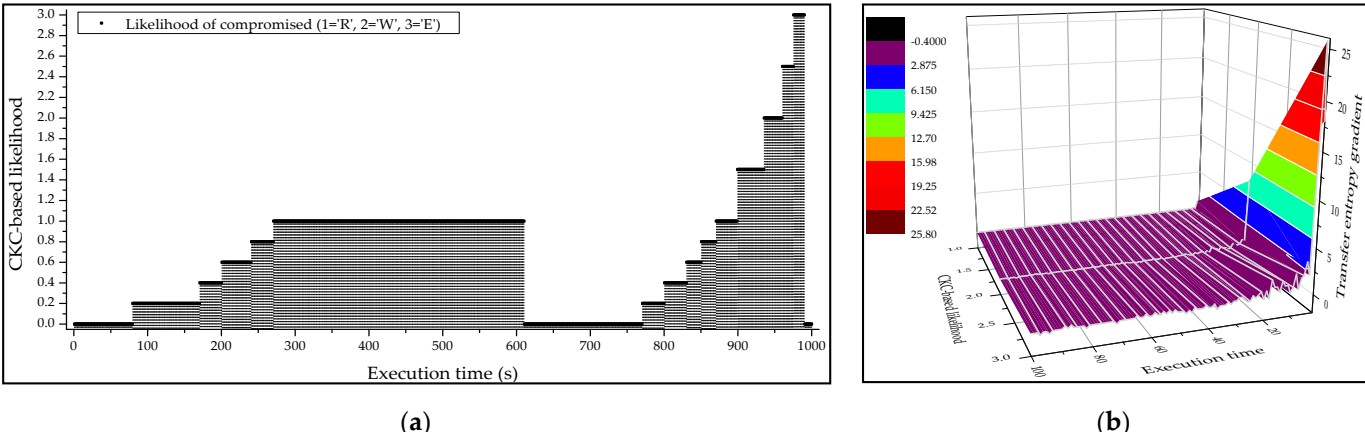

(**a**) (**b**)

**Figure 10.** CKC-based compromise likelihood and multi-dimensional entropy-based results in the IODA simulation. (**a**) CKC phases based on reconnaissance, weaponization and exploitation, (**b**) transfer entropy-based operational surfaces.

Figure 10a defines the scale of deception efficiency as the step-by-step occupation probability by standardizing the optimized LPC-MTD and HS-Decoy-based deception sequences with CKC simulation, and beaconizing the final compromise points in the CKC phase. First, in the reconnaissance phase, an occupation success rate of 60% was derived from an execution time exceeding 201 s, and 100% was calculated from 271 s to achieve the initial infiltration and occupation. After that, while collecting lower layer information sequentially, such as hosts and services, based on the defender's random trait information used for the initial invasion up to 610 s, chain attacks continue to fail until the 770 s mark, because the validity of the defender trait that was utilized by the attacker had expired due to the mutation scheme after the immediate intervention of the defender, who identified that the final compromise point surfaced as a reconnaissance phase was occupied. At this time, the defender's response was late when occupying the reconnaissance phase, because it performed loose mutations in LPC-MTD and intentional HS-Decoy leakage. The invaded environment was a false beacon that was constructed by the defender, so no security issues existed. However, prior to the defender's late intervention and mutation, the attacker already collected and enumerated additional information to continue the new attack chain, enabling it to occupy quickly while drawing a sharp upward gradient. In other words, from 871 s, the attacker re-occupies the reconnaissance phase and completes invasion to the weaponization and exploitation phases at 936 and 976 s, respectively. Subsequently, based on the SDN controller's rule-based control that detected the attacker payload that continues from the exploitation phase, it shows that the attacker's asymmetric superiority relationship was completely resolved under the defender's intention. Accordingly, Figure 10b shows that the attacker's asymmetric superiority and the effectiveness of the defender's traits that were collected at a specific point in time declined and expired sequentially based on transfer entropy under the deceptive intention of the defender who had identified a random attacker pursuant to (a)'s result.

## 5. Discussion

This study proposed an organizational OSINT-based HS-Decoy and an LPC-MTD for artificial information exposure to improve and optimize the operational efficiency of cyber deception applied in an organizational operating environment. This made it possible to produce theoretical diversity that alleviates the operational limitations in the actual organizational environment created by existing MTD and decoys. Summarizing the results, when formulated the attack success probability, artificial information exposure

ratio, deceptive probability, and enticingness probability as major comparative metrics, and the total attack efficiency through LPC-MTD and OSINT-based HS-Decoy was reduced by 287%, on average. In addition, it was verified that the deception efficiency can also be improved by 382%, on average. Subsequently, as a result of detailed parameter-based sensitivity analysis for optimization of proposed deception concepts, the optimized periodic mutation time slot was calculated as 174s, the optimized mutation scope was 2048, mutation selective batch pool size was 169, and exploit or dummy packet size was 2048 bytes. Moreover, it was proved that the increase rate of deception overhead by an organization for U.S organizational environments simulated as standardized OSINT information groups based on third party archives and related services was also reduced by 174% on average, and the LPC-MTD-based each trade-off values was also calculated.

However, the limitations and related improvement plan are summarized, as follows:

- Reliability and practicality: to prevent legal disputes and comply with all research ethics, the presented comparative experiment results were constructed in a test bed using only public OSINT information that was provided by third parties as a public service. Because the experiment was conducted within a limited range of OSINT, the results of organizational deception in an actual environment may differ from those that were obtained in our experiment. In addition, since insertions and updates of OSINTs for major organizations within OSINT standardization modules have been performed for a long time, the availability problem will also remain in outlier adjustment and error reduction.

- Scalability and operability: the proposed LPC-MTD and HS-Decoy-based applied deception research is based on the hierarchical OSINT complex module and CVE vulnerability module based on the social engineering OSINT information group. It only focused on the design, analysis, and optimization of a defendant superior information exposure type deception environment specialized in organizational traits, so domain scalability issues are likely to occur. Thus, it would be necessary to construct a formula and pseudo code unique to each organization, while also verifying the quantitative performance based on the organizational-specific attack-defense scenarios. Considerable targets include operating environments that are based on US Military and security standards. In addition, the OSINT and vulnerability DB for information exposure will also be customized according to the operating standards.

- Strategicization: to advance the applied deception model in rapidly changing attack-defense scenarios and improve performance, the game theory strategy used in previous studies should be applied from the perspective of non-cooperative attack–defense competition relationships. To this end, it is necessary to consider the following: PBE-based Nash equilibrium, Stackelberg competition, asymmetric superiority and inferiority coordination signal strategy between the attacker and defender, Bayesian persuasion, FLIPT-based defender's strategy, and deceptive SIR. Moreover, the diversification of the organization surface to which the improved HS-Decoy and LPC-MTD were applied should also be performed quantitatively.

- Surfaceization: because HS-Decoy and LPC-MTD have very high weights on the knowledge-based graph-based social engineering OSINT, a hierarchical weight subtraction and addition scheme will be needed according to the parameter trait in the OSINT. For this purpose, graph-based correlation analysis and information entropy analysis can also be considered. In addition, it is necessary to consider deception using detailed scenarios according to organizational goals as well as deception efficiency verification and optimization for a simple organizational operating environment. Therefore, the attack metrics in the CVE-based attack graph for the sensitivity analysis of each attack vector should be classified, as shown in Figures A3 and A4.

## 6. Conclusions

To improve and optimize the deception operation efficiency in the organizational environment, this study proposed an organizational OSINT-based HS-Decoy and a com-

petitive exposure-based LPC-MTD. We also presented an integrated deception architecture combining the proposed concepts, multilayered processes, metrics, and variables. The experiments showed that the attack efficiency was 2.8 times less than that of existing MTD and decoys, while the artificial deception efficiency of the defender's superiority was 3.8 times better. We could also optimize the trade-off threshold while reducing the rate of increase of organizational overhead by 1.7 times. Consequently, we could derive the deceptive utilization of the OSINT concept, with LPC-MTD mutation cycle selectivity and error attenuation, and the practical reliability and operability improvement through the optimization of organizational HS-Decoy. In order to increase the operational reliability of the established HS-Decoy and LPC-MTD-based combined model and diversify it based on detailed organizational goals, we advanced it as a number of game strategies that were based on international US Military standards and security standards.

**Author Contributions:** Conceptualization, S.S. and D.K.; methodology, S.S.; software, S.S.; validation, S.S. and D.K.; formal analysis, S.S. and D.K.; investigation, S.S.; resources, S.S.; data curation, S.S.; writing—original draft preparation, S.S. and D.K.; writing—review and editing, S.S. and D.K.; visualization, S.S.; supervision, D.K.; project administration, S.S. and D.K; funding acquisition, D.K. All authors have read and agreed to the published version of the manuscript.

**Funding:** This research received no external funding.

**Institutional Review Board Statement:** Not applicable.

**Informed Consent Statement:** Not applicable.

**Acknowledgments:** This work was supported by a Kyonggi University Research Grant (2020).

**Conflicts of Interest:** The authors declare no conflict of interest.

## Appendix A. Supplementary Data

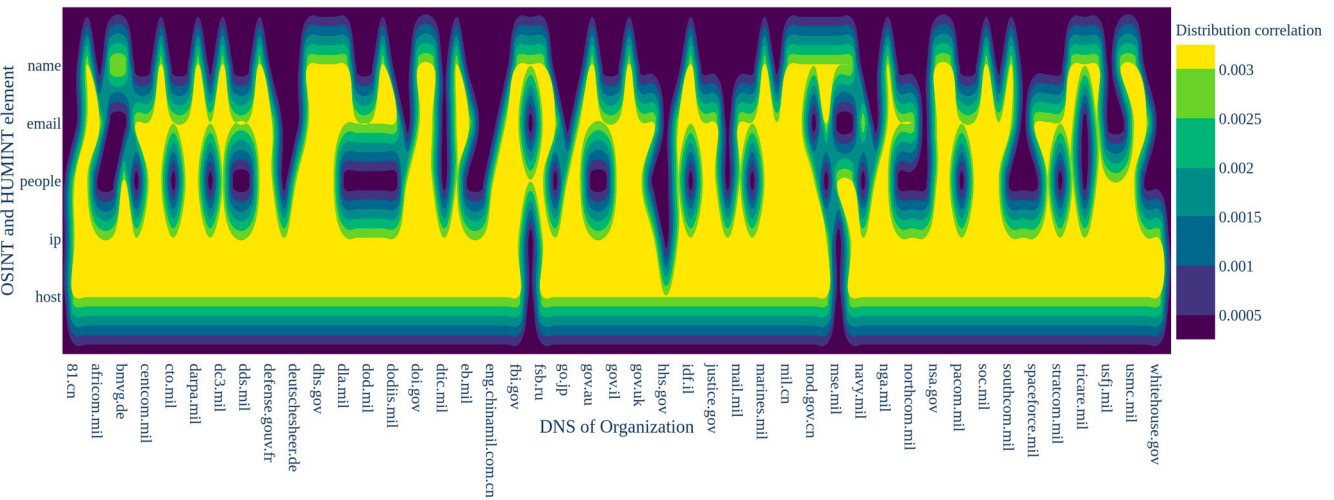

**Figure A1.** Relative entropy-based probability distribution correlation of organizational OSINT with Table A1.

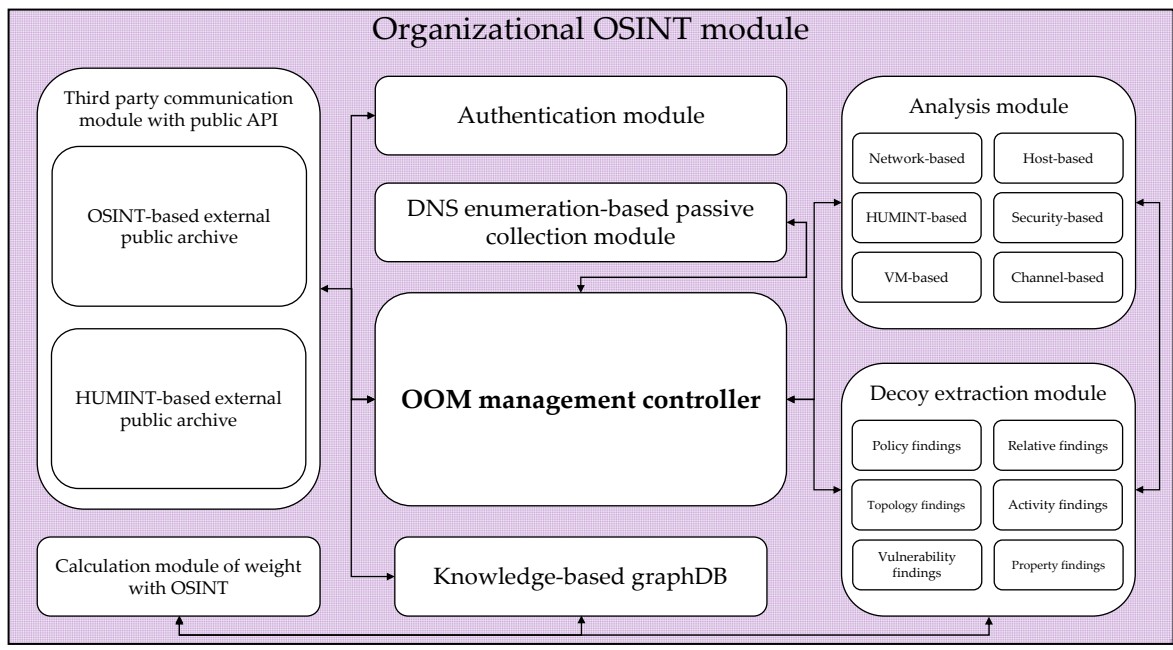

**Figure A2.** GraphDB-based organizational OSINT module in Figure 4.

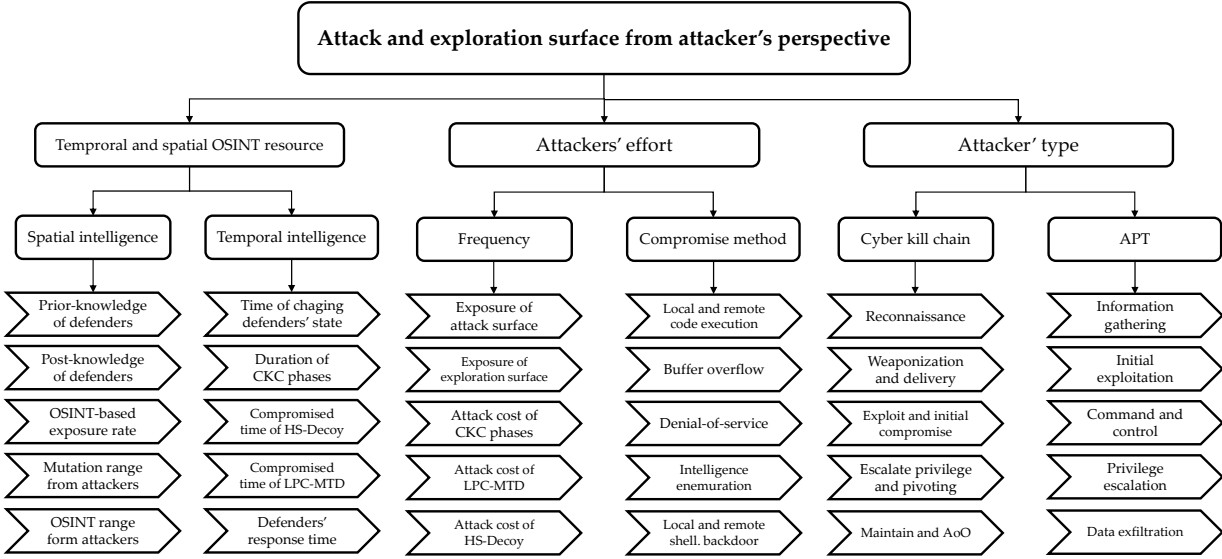

**Figure A3.** Hierarchical compound metric tree based on attackers' perspective.

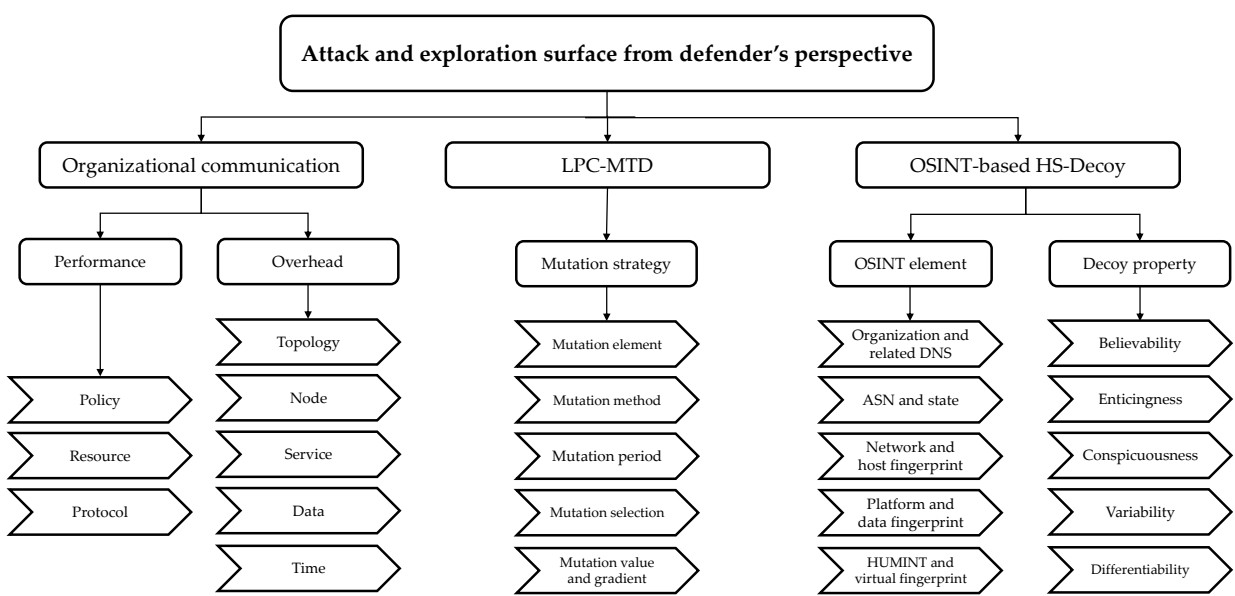

**Figure A4.** Hierarchical compound metric tree based on defenders' perspective.

**Table A1.** Overview of OSINT composition with DNS enumeration for LPC-MTD and HS-Decoy.

| OSINT Element in HS-Decoy | Description and Examples |
|---|---|
| DNS based on TLD and SLD | Hierarchical DNS-based organizational intelligence (e.g., army.mil, mail.mil, pacom.mil, gov.uk, gc.ca) |
| Country | Country information by organization |
| ASN | Autonomous system number by organization (e.g., AS721, AS306, AS385, AS16509) |
| Network topology and infrastructure | Overall public network infrastructure by organization (e.g., firewall-DMZ-NAC-SDN-Cloud-host-OS structure) |
| Gateway and router-based network device | Detailed network device identifier by organization (e.g., CISCO's branch, Juiper's access point) |
| IPv4 and IPv6 address | Public or private IPv4/IPv6 with prifix and subnet (e.g., 29.0.0.0/8, 143.140.0.0/14, 2605:3100:fffb:/48) |
| Multi-layered host system | Connected device-based information in infrastructure (e.g., application server, database, local file server, archive and backup server) |
| Port | Inbound/outbound port socket-based information (e.g., well-known, registered, and dynamic on 0~65535) |
| OS | OS fingerprint-based information by each host (e.g., HP-UX, RHEL 7, and Windows IIS 2016) |
| Protocol-based service and crontabs | Service and related protocol-based crontab information (e.g., FTP, SSH, SMTP, DNS, HTTPS, RPC, NetBios) |
| E-mail | E-mail-based OSINT and HUMINT by employee (e.g., xxx@hotline.aa, yyy@vpn.bb.cc, zzz@cloud.dd) |
| Organizational employee-based HUMINT | Detailed HUMINT with Facebook and LinkedIn, Twitter |
| Virtual machine | Virtual host-based information in cloud or virtual nets |
| VPN | VPN and proxy channel-based information (e.g., vpn13.aaa.bbb, vpn_mil.cc.dd) |
| Cloud environment | Single cloud or federated cloud-based information (e.g., cloud133.aa.bb, semi-isolated_cloud.ccc.ddd) |
| DMZ | DMZ server, farm, and related limited network zone |
| NAC | NAC server, topology, and related access control policy |
| Hotline channel | Private hotline communication-based OSINT (e.g., hotline445.aaa.bbb, com.hotline.cc.dd) |
| Security solution | Firewall and sandbox-based security information (e.g., ipfire.xxx.yyy, sandbox.host114.aa.bb) |

**Table A2.** Examples of deceptive parameters of LPC-MTD.

| Parameter | Value | Description |
|---|---|---|
| Time slot length for periodic mutation (s) | 1–100,000 | Mutation time slot to limit attackers' effectiveness |
| Number of surface views by attackers | 0–50 | Number of surface views that attackers can see at a mutation states for artificial disinformation of organizational OSINTs at a specific attack time |
| Number of surface views by defenders | 10–100 | Number of surface views that defenders can see at a mutation states for monitoring and verification of OSINT-based disinformation at a specific attack time |
| Mutation range of IP-based OSINTs | $2^8$–$2^{36}$ | Mutation range of IPv4/IPv6 pool and subnet-based OSINTs |
| Mutation range of port-based OSINTs | $2^{10}$–$2^{16}$ | Mutation range of port and service socket-based OSINTs |
| Mutation range of host-based OSINTs | 1–100 | Mutation range of platform and host node-based OSINTs |
| Mutation range of CVE-based vulnerabilities | 10–50 | Mutation range of vulnerabilities in HS-Decoy |
| Mutation range of packet data-based OSINTs | $2^0$–$2^{11}$ | Mutation range of payload and header size-based OSINTs |
| Mutation range of HUMINTs | 0–20 | Mutation range of personal information-based HUMINTs |
| Mutation range of VM-based OSINTs | 0–20 | Mutation range of virtual node and channel-based OSINTs |
| Mutation range of security-based OSINTs | 0–3 | Mutation range of security solution-based OSINTs |
| Enabling mutation selection weight of OSINT | 0.01–0.70 | Bayesian-based dynamic batch selection probability for each OSINT-based element's LPC-MTD strength |
| Enabling mutation gradient of OSINT | 0.01–0.90 | LPC-MTD gradient to reduce unnecessary overhead |
| Enabling aperiodic mutation | True | Mutation for immediate intervention action when attackers' initial compromise is detected |
| Enabling MAC address shuffling | False | Enabling shuffling virtual MAC address mapped with network fingerprint based on virtual pairs for enhancing security although there are penalties when performing migration |
| Enabling CKC-based attack chain scheme | True | Enabling scenario-based attack chain for calculation of CKC-based propagation impact in LPC-MTD |
| Enabling true random generator of LPC-MTD | True | Enabling true random generator-based sampling scheme for compound Monte Carlo simulation |

**Table A3.** Examples of deceptive parameters of OSINT-based HS-Decoy.

| Parameter | Value | Description |
|---|---|---|
| Number of organizational intelligences | 1–90 | Number of OSINT-based organizational intelligence |
| Number of network topologies and infrastructures | 1–5 | Number of network topologies by organization |
| Number of communication nodes | 10–100 | Number of participating servers or hosts in topology |
| Number of shutdown system nodes | 0–30 | Number of down nodes at a specific mutation time |
| Number live system nodes | 10–100 | Number of live nodes at a specific mutation time |
| Number of OS platforms | 1–5 | Number of operating systems by host |
| Number of services | 1–20 | Number of crontab or protocol-based services by OS |
| Number of vulnerabilities | 10–50 | Number of vulnerabilities based on CVE |
| Level of vulnerabilities | L-M-H-V-U-C | Rating level for vulnerabilities with CVSS score |
| Number of HUMINT intelligence | 0–20 | Number of personal intelligence-based HUMINTs |
| Number of virtual environments | 0–20 | Number of VM-based virtual environments |
| Number of multitenancy clusters | 0–5 | Number of cloud, DMZ, and hotline-based clusters |
| Number of security solutions | 0–3 | Number of firewall, NAC, sandbox-based solutions |
| Enabling enticingness of HS-Decoy | 0.3–1.0 | Enabling in-depth decoy process to attackers initially induced through artificial disinformation behavior |
| Enabling conspicuousness of HS-Decoy | 0.7–1.0 | Enabling protrusion scheme of arbitrary defenders' OSINT within the attack and exploration surfaces |
| Enabling variability of HS-Decoy | 0–1 | Enabling diversity of OSINT with batch sampling to reveal the characteristics of each organization |

**Table A4.** Configuration of communication and simulation parameters in SDN-based testbeds.

| Parameter | Value | Description |
|---|---|---|
| Packet size (byte) | 16–1500 | Calculating various protocol-based packets in SDN |
| PER | 0.0–0.1 | Packet error rate by protocol in SDN |
| BER | 0.0–0.1 | Bit error rate by protocol in SDN |
| Minimum security length (bit) | 112 (NIST), 224 (BSA), 256 (NSA) | Security strength for organizational infrastructures |
| Number of LM CKC phases | 4–7 | Attackers' phases with Lockheed Martin cyber kill chain when related CKC-based attack metrics are set |
| Number of surface views allowed by the attackers | Declared according to LPC-MTD and HS-Decoy's variables | Maximum number of surface views attackers can see at a specific CKC phase or mutation time |
| Number of surface views allowed by the defenders | Declared according to LPC-MTD and HS-Decoy's variables | Maximum number of surface views defenders can see at a specific mutation time for proactive defense |
| Number of scenarios | 1–4 | Number of attack and defense chain-based scenarios |
| Maximum attack time (s) | 172,800 (2 days) | Overall attack time for initial compromise by attacker |
| Maximum defense time (s) | 3600 (1 h) | Overall defense time to reduction of surfaces |
| Maximum simulation time (s) | 259,200 (3 days) | Overall operation time by each OSINT elements, LPC-MTD, and HS-Decoy metrics |

**Table A5.** Configuration of experimental elements in SDN-based testbeds and MATLAB-based post-processing.

| Element | Value |
|---|---|
| Operating system | Ubuntu 20.04 LTS (VM) |
| CPU 1 and GPU 1 for simulation in SDN | INTEL I7-10700, NVIDIA GTX 2060 super |
| CPU 2 and GPU 2 for mathematical post-processing | INTEL I7-10875H, NVIDIA GTX 2060 |
| Type of communication nodes | Producer, controller, forwarder, interpreter, consumer, and filter |
| Enabling filter nodes for deception | True |
| Maximum number of re-transmissions count | 0–3 (Dynamic declaration according to the number of SDN switches) |
| Maximum number of re-synchronization count | 0–3 (Dynamic declaration according to LPC-MTD mutation) |
| SDN protocol | OpenFlow 1.4, OpenFlow 1.3 |
| SDN controller | ONOS Sparrow LTS 2.2.0, Ryu |
| OF-based SDN virtual switch | Open vSwitch LTS 2.10.1 |
| Attack scenario modeling for assessment in SDN | LM CKC and Mandiant APT with CVE and CAPEC |
| Number of IODA-based containers in SDN | 1–10 |
| Number of virtual switches in SDN | 3–8 |
| Number of vIP pairs with one rIP | 1–10 |
| Number of vPort pair with one rPort for service | 10–1023 |
| Maximum routing rule for mutation | $2$–$2^{10}$ |
| Targeted protocol model applied within SDN | TCP/IP model |
| Integrated SDN environment referenced [60] | SDN Hub VM |
| Major open source software referenced [61–65] | T-Pot, DejaVU, Cuckoo Sandbox, VMCloak, OpenCTI, Amass, theHarvester, OSINT Framework, Recon-ng, Social Engineer Toolkit |
| Third party archive and related API services [66] | Declared according to OSINT-based open source software(e.g., Bing, Duckduckgo, HackerOne, RapidDNS, AlienVault, Censys, Cloudflare, BufferOver, VirusTotal, Facebook, Twitter, Shodan) |

**Table A6.** Optimized trade-off results based on deception simulation by major US organization in Figure 9.

| Category | US Organization | Trade-Off Based on Time Slot Length (s) |
|---|---|---|
| Armed forces | US Army | 23 |
| | US Air Force | 19 |
| | US Navy | 26 |
| | US Marine Corps | 25 |
| | US Coast Guard | 21 |
| | US Space Force | 2 |
| Unified combatant command | USPACOM | 13 |
| | USEUCOM | 6 |
| | USSOUTHCOM | 7 |
| | USCENTCOM | 13 |
| | USNORTHCOM | 6 |
| | USCYBERCOM | 5 |
| | USSTRATCOM | 8 |
| | USSPACECOM | 5 |
| Intelligence community | US CIA | 20 |
| | US NSA | 19 |
| | US FBI | 28 |

**Table A7.** List of annotations for LPC-MTD and HS-Decoy equations in this paper.

| Annotation | Description |
|---|---|
| $s$ | selected OSINT element for initial exposure of artificial information |
| $S$ | set of OSINT elements for initial exposure of artificial information |
| $\mu(s)$ | predictability of the defender's secret given exposed information in the past |
| $l$ | selected perturbation-based OSINT element with defender superiority |
| $\bar{l}$ | maximum number of defender's perturbation-based OSINT for leakage |
| $\underline{l}$ | minimum number of defender's perturbation-based OSINT for leakage |
| $L$ | set of perturbation-based OSINT elements with defender superiority |
| $LC$ | overall leakage cost in testbed |
| $T$ | set of mutation time slot length sequence |
| $i$ | the $i$th mutation time slot length sequence in $T$ |
| $M$ | set of attack and exploration surfaces generated through mutation |
| $m_i$ | selected attack and exploration surface $i$ generated through mutation |
| $\widetilde{m}_{ij}$ | probabilistic difference between attack and exploration surface $i$ and $j$ |
| $\tau_i$ | defender's response time at the current mutation time slot length from surface $i$ |
| $\bar{\tau}$ | defender's maximum response time |
| $\underline{\tau}$ | defender's minimum response time |
| $C$ | overall mutation cost in testbed |
| $C_{ij}$ | mutation cost from surface $i$ to $j$ |
| $\theta$ | overall performance delay time in testbed |
| $O$ | set of organizational OSINT elements |
| $o$ | selected OSINT element |
| $H$ | set of HS-Decoy |
| $h$ | selected HS-Decoy |
| $h'$ | another HS-Decoy |
| $D$ | defender in testbed |
| $A$ | attacker in testbed |
| $P$ | preference of HS-Decoy |
| V | view of surface with attacker and defender |
| $\delta$ | conspicuous OSINT element |
| CD | compromise detection |
| $\varepsilon$ | detection threshold of HS-Decoy |
| CT | contiguity with HS-Decoy |

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
