# Peer review of "OSINT-Based LPC-MTD and HS-Decoy for Organizational Defensive Deception"

_applsci, doi:10.3390/app11083402_

Round 1

Reviewer 1 Report

The paper contribution seems interesting and significant. Overall the paper reads well, although it is quite dense, which authors try to mitigate with figures. I provide only two critiques to the document. First, some of the above mentioned are too low resolution to the point they are unreadable. Some are quite important to understanding the scenario, such as Figure 6. Figure A1 suffers from the same problem. Such issues can be easily managed.

Second critique concerns the SDN setup and validation. The authors fail to explicitly describe the setup which prevents scrutiny and replications of results. What software was used, what was the setup, on what hardware, which attack modes were tested, results seem to provide an aggregation of those and this must be clarified), which scenarios... all of these important questions must be addressed for the above mentioned reasons in a comprehensible fashion. 

Minor typos: line 540 - displaced equal sign =

Author Response

Dear reviewer,

Thank you, your comments on our paper “OSINT-based LPC-MTD and HS-Decoy for Organizational Defensive Deception”. 

I modified our prior manuscript (Manuscript ID: applsci-1144754).

Please check my rebuttal comments in the attached file.

Thank you very much.

The Authors.

Reviewer 2 Report

The authors proposed a stydy abouth existing moving target defense (MTD) and decoy concepts, also presenting concepts about OSINT-based hiearchical social engineering decoy strategy considering the actual fingerprint of each organization.

The proposed study is interesting but there are some points that the authors should better discuss.

The authors should be better described the novelties of their study with respect to existing ones, also underlying the main pros and cons of the examined approaches.

Furthermore, the authors should provide more details and discussion about the obtained results. The Discussion section also needs to be improved by analyzing the outcome of evaluation section.

I suggest to further analyze more recent approaches about the examined topics. In particular, I suggest the following papers to further investigate more recent deception approaches:

1) FORGE: a fake online repository generation engine for cyber deception. IEEE Transactions on Dependable and Secure Computing.

2) Generating Fake Documents using Probabilistic Logic Graphs. IEEE Transactions on Dependable and Secure Computing.

Finally, I suggest to perform a linguistic revision.

Author Response

(The authors gave the same response as above.)

Round 2

Reviewer 2 Report

I think that the authors have addressed all my concerns.